# Solving for ambiguities in radar geophysical exploration of planetary bodies by mimicking bats echolocation

Leonardo Carrer [1] & Lorenzo Bruzzone [1]

Sounders are spaceborne radars which are widely employed for geophysical exploration of celestial bodies around the solar system. They provide unique information regarding the subsurface structure and composition of planets and their moons. The acquired data are often affected by unwanted artifacts, which hinder the data interpretation conducted by geophysicists. Bats possess a remarkable ability in discriminating between a prey, such as a quick-moving insect, and unwanted clutter (e.g., foliage) by effectively employing their bio-sonar perfected in million years of evolution. Striking analogies occur between the characteristics of bats sonar and the one of a radar sounder. Here we propose an adaptation of the unique bat clutter discrimination capability to radar sounding by devising a novel clutter detection model. The proposed bio-inspired strategy proves its effectiveness on Mars experimental data and paves the way for a new generation of sounders which eases the data interpretation by planetary scientists.

---

[1] Department of Information Engineering and Computer Science, University of Trento, Trento 38123, Italy. Correspondence and requests for materials should be addressed to L.B. (email: lorenzo.bruzzone@unitn.it)

Radar sounders are spaceborne sensors, which exploit the interaction between electromagnetic waves and matter to probe the subsurface of celestial bodies. Bats are unique animals whose million years evolution has resulted in the refinement of a bio-sonar, which is fundamental for hunting preys. Their bio-sonar performances are still unmatched by human-made radar and sonar systems.

These two apparently different fields of science sharing ties with electromagnetism for planetary exploration and biology offer inspiration to open up new ways of dealing with ambiguities in radar geophysical exploration of planetary bodies. In order to investigate the subsurface of a given celestial target, a planetary radar sounder transmits pulsed electromagnetic radiation and, as it travels through the subsurface, each dielectric discontinuity in the ground material results in part of the signal being reflected toward it. These signal reflections are subsequently recorded by the sensor thus forming an echo trace for each given acquisition (Fig. 1). By analyzing these echo traces, it is possible to obtain crucial information on the subsurface structure and composition. By stacking together subsequent echo traces a bidimensional image (i.e., radargram) is formed where one axis represents depth and the other one the position of the sensor along its ground track. Radar sounders have been employed to probe the subsurface of Mars[1,2] and the Moon[3,4]. As an example, by using radar sounder data, scientists were able to confirm the presence of nearly pure water ice within the South Polar Layer Deposits of Mars[5] and to observe its accumulation and erosion[6].

When operating, planetary sounders antennas (which are assumed to be dipoles due to mechanical reasons) are always pointed toward nadir direction with respect to the surface and illuminate large surface and subsurface regions. The very large antenna footprint implies that off-nadir surface reflections (i.e., surface clutter) of the transmitted signal can be disguised for echoes coming from the subsurface. This causes a serious issue in terms of data interpretation (Fig. 2). The clutter main driving factors are the surface characteristics such as roughness and electromagnetic backscattering properties. The most popular approach to clutter identification is to simulate the ambiguous echo signals produced by the surface and then compare them

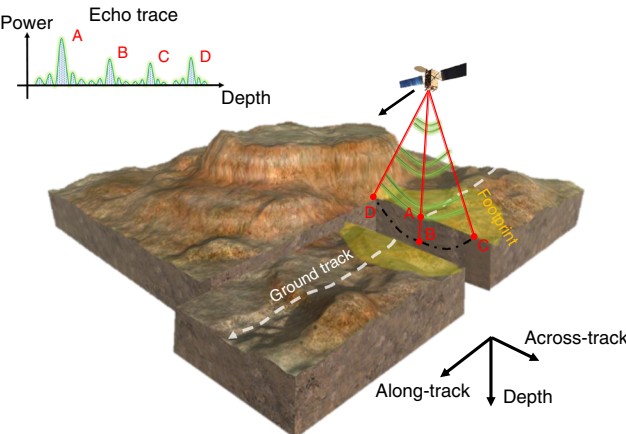

**Fig. 1** Radar sounding acquisition geometry. The radar sounding main goal is to acquire the signal coming from the nadir surface and subsurface regions (points A and B). Due to the large antenna footprint, the echo trace contains unwanted reflections coming from the antenna footprint sides (points C and D). A radar sensor measures time intervals occurring between the signal transmission and reception and not absolute distances. This implies that reflections from points C and D can be disguised as subsurface signal. In fact, in a radar system distance is inferred from time delay by knowing the wave speed in the medium

with the experimental data[7]. This approach requires the availability of a digital elevation model of the surface, which should be acquired by other instruments (e.g., a laser altimeter), with a sufficient resolution to give accurate results. Unfortunately, very often a surface digital elevation model is not available or it has an insufficient resolution with respect to the radar wavelength.

Different attempts have been made to find a distinctive domain (i.e., an electromagnetic property of the signal), where disambiguation between clutter and subsurface signal can be performed. Recent papers proposed to detect clutter by focusing on the antenna pattern[8], the polarization[9], and interferometric diversity[10]. The approach based on antenna pattern diversity consists in deploying two different antennas. The primary antenna points toward nadir direction, whereas the secondary antenna illuminates surface regions to the sides of the main one. The signal received by the secondary antenna is assumed to contain only off-nadir surface clutter and it is subtracted from the signal acquired by the primary antenna thus achieving clutter reduction. Polarization can be exploited for solving clutter ambiguities by transmitting circularly polarized waves. Nadir and subsurface reflections invert the sense of polarization once, whereas the off-nadir clutter reflections result in a double bounce. By projecting the received signal onto a suitable feature space, it is in principle feasible to distinguish the off-nadir reflections from the subsurface ones. The interferometric strategy consists in measuring the phase difference between the echoes received by two spatially separated antennas. The expected phase distribution for the nadir subsurface echoes differs from the one of the clutter thus making discrimination possible. However, none of the aforementioned strategies has proven to be the final solution to this specific problem. Moreover, the methods based on polarization and interferometry have never been tested on actual experimental data acquired from space.

Different animal species, such as dolphins and bats, rely on echolocation for foraging and navigation[11]. Their bio-sonars characteristics and processing scheme share many similarities with the strategies being adopted in radar sounders design and operations. As an example, some bat species transmit linear frequency-modulated signals[12] and adapt the time between each signal transmission according to the target distance[13]. When hunting preys, bats usually fly in complex environments such as canopies of forests. This suggests that their sonar system should be capable of dealing with unwanted echoes coming from the surroundings[14–16]. Moreover, big brown bats need to face a given prey with their mouth. This implies that their targets will always be oriented toward nadir direction. It is therefore clear that a radar sounder and a big brown bat share a similar acquisition geometry (Fig. 3) even if in very different scenarios. Bates et al.[14] made a major step forward in unveiling the processing scheme of big brown bats (*Eptesicus Fuscus*) that is associated with their remarkable clutter mitigation performance. The main concept behind the *Eptesicus Fuscus* clutter reduction technique is to exploit frequency diversity. Big brown bats modulate two different harmonics over the same linear frequency-modulated signal. The pattern beam-width (i.e., spatial distribution of the transmitted energy) is frequency dependent and narrower for the higher harmonic with respect to the fundamental one. Moreover, the signal attenuation due to atmospheric effects is frequency dependent too. By performing the ratio of the echo power between the two harmonics, big brown bats can predict the echo direction of arrival and range.

Here we successfully adapt and implement the big brown bats clutter mitigation mechanism to radar geophysical exploration of planetary bodies. We propose a clutter detection model inspired by the bats processing strategy and tailored to the specific case of radar subsurface sounding. This results in a model that provides

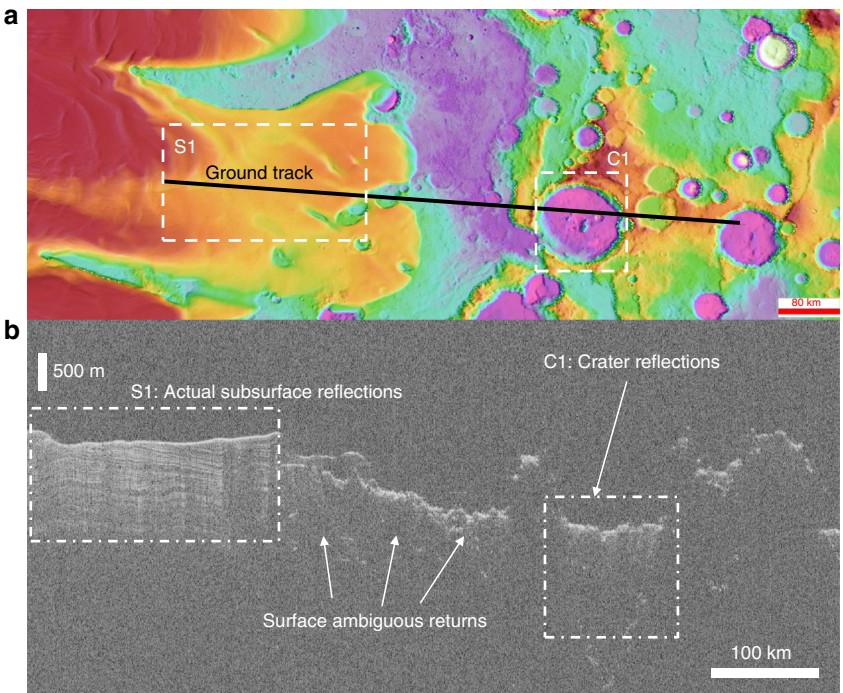

**Fig. 2** Data interpretation ambiguities due to clutter. **a** Ground track of the SHallow RADar[2] acquisition 241301 projected over Mars South Pole region of Promethei Lingula. S1 represents a region where the instrument detects the subsurface-layered deposits, while C1 is an example of ambiguous return due to off-nadir reflections from the crater rim. **b** SHARAD radargram of acquisition 241301. From the image, we can notice that there are many ambiguous returns, which cause difficulties in the data interpretation

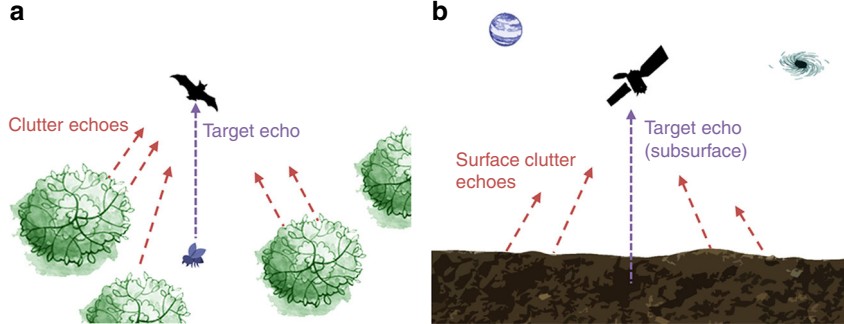

**Fig. 3** Bats and radar sounder similarity in acquisition geometry. Similarity between **a** bats and **b** radar sounders acquisition geometry. In the bat case, side clutter echoes are coming from foliage, while in the radar sounder case are coming from surface features. For both cases, the target of interested is always located in nadir direction

simple physical conditions for which clutter ambiguities can be resolved. We then apply the proposed bio-inspired model to experimental data acquired over different regions of Mars to assess the effectiveness of the presented approach.

## Results

**Physical analogies between bats bio-sonar and radar sounding.** In this section, we analyze the *Eptesicus Fuscus* clutter mitigation strategy to find and study the difference and analogies with radar sounding. We argue that it is possible to adapt the bat clutter mitigation technique to the radar sounding case. The main obvious difference is that bats emit ultrasound waves while radars electromagnetic (EM) waves. Nevertheless, the two domains share many similarities such as interference, diffraction, and refraction phenomena. A clear analogy, which is very relevant for this work, is the one between the bats sonar equation and the sounder radar equation. With reference to the big brown bat, the sonar equation that relates the echo power received (denoted as $P_{b,n}$) as function

of the off-nadir angle $\theta$ and $n$-th harmonic $f_n$, $n = 1, 2$, is as follows:[17]

$$P_{b,n}(\theta) = \frac{P_{call}\, G_{tr}(\theta, f_n)\, A_{ear}\, \sigma_b^0(\theta, f_n)\, A_t\, e^{-2\alpha_b(f_n)d}}{(4\pi)^2 d^4}, \quad (1)$$

where $P_{call}$ is the power of the sonar call transmitted by the bat, $d$ is the target (e.g., insect) distance from the bat and $G_{tr}(\theta, f_n)$ is the gain of the transmitting antenna. It depends on the shape of the bats nose or mouth. $A_{ear}$ is the area of the receiving antenna (i.e., bat's ear) and $\sigma_b^0(\theta, f_n)$ is the acoustic cross-section of the target. It indicates the target (e.g., a moth) capability of reflecting the transmitted energy back to the bat per unit area. This quantity depends on factors such as the shape and material of the target. $A_t$ is the target area, a large target reflects more energy back to the bat. Finally, $\alpha_b(f_n)$ is the atmospheric attenuation constant, which greatly affects the amount of echo power received by the bat and

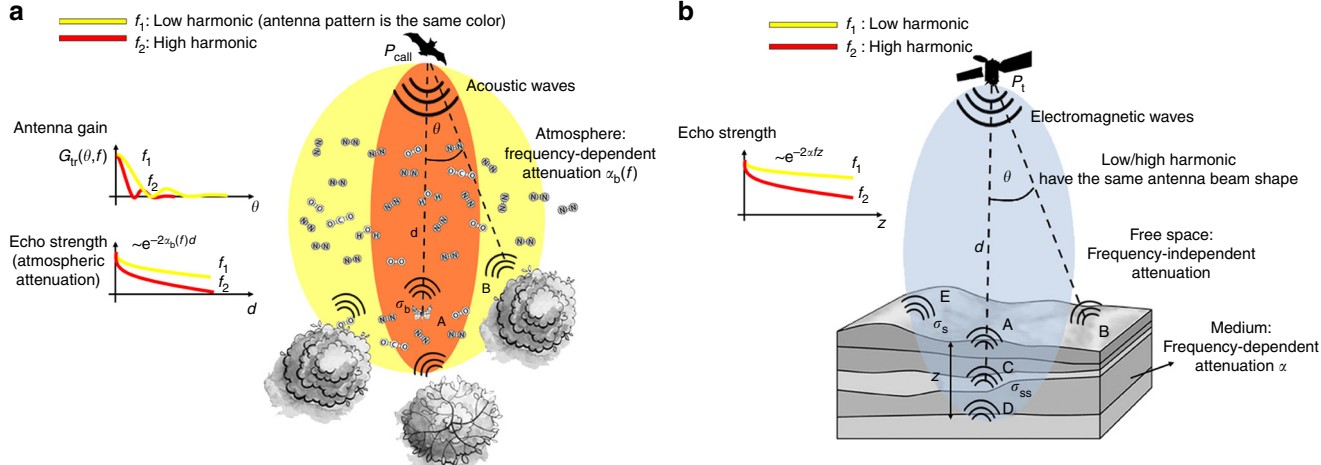

**Fig. 4** Bats and radar sounder acquisition geometry scheme. **a** Visualization of bats acquisition geometry assuming multi-harmonic signal transmission. The bats exploits a combination of atmospheric attenuation and antenna pattern diversity to distinguish clutter echoes coming from foliage. **b** Acquisition geometry for radar sounders assuming multi-harmonic transmission. The echo strength reduction in the subsurface depends on the transmitted frequency

sensibly increases with frequency[18]. The relevant parameters of the above equation are sketched in Fig. 4a.

The bats multi-harmonic scheme is reflected by modeling in the radar equation the transmission being performed at two different harmonics namely $f_1$ and $f_2$. Both harmonics modulate the same chirp signal (i.e., linear frequency modulated) with equal bandwidth and pulse width. This is consistent with the bats transmission scheme presented in Bates et al.[14]. The echo power received from the surface at a given off-nadir angle $\theta$ and at the $n$-th harmonic ($n = 1$, 2) harmonic $f_n$ is given by the radar equation[19] and it is equal to:

$$P_{s,n}(\theta) = \frac{P_t\, G(\theta, f_n)\, A_e\, \sigma_s^0(\theta, f_n) A_{ill}}{(4\pi)^2 d^4},\qquad(2)$$

where $P_t$ is the radar transmitted power, $G(\theta, f_n)$ is the antenna gain, $A_e = G(\theta, f_n)c^2/(4\pi f_n^2)$ is the antenna aperture, where $c$ is the speed of the light, $\sigma_s^0(\theta, f_n)$ is the surface backscattering coefficient, $A_{ill}$ is the illuminated area and $d$ is the distance between the radar antenna and an arbitrary point of the surface. The surface backscattering coefficient measures the ability of a given surface to reflect electromagnetic energy. The forementioned quantities are visually described in Fig. 4b. By comparing Eqs. (1) and (2), it is clear that the resulting echo power in the two cases depends on similar quantities. The main difference is that in the radar case, there is no exponential attenuation of the echo power due to the atmospheric effect as in the bat case. This is expected because in planetary radar sounder, the medium between the radar antenna and the surface is assumed to be vacuum. However, as stated in the introduction, the main goal of a radar sounder is to investigate the subsurface rather than the surface. In this case, the echo power at the $n$-th harmonic is described by:

$$P_{ss,n}(\theta) = \frac{P_t\, G(\theta, f_n)\, A_e\, \sigma_{ss}^0(\theta, f_n)\, A_{ill}}{(4\pi)^2 (d+z)^4}\left(1 - R_{01}^2(0)\right)^2 e^{-2f_n \alpha z},\quad(3)$$

where $\alpha$ is the subsurface two-way attenuation factor, $z$ is the depth into the ground, $R_{01}$ is the Fresnel reflection coefficient between the first medium and the second one, $\sigma_{ss}^0(\theta, f_n)$ is the subsurface backscattering coefficient and $z$ is the ground penetration depth. The subsurface attenuation is function of the ground material electrical conductivity and relative permittivity. The Fresnel reflection coefficient between the $i$-th and $j$-th

medium is expressed as:

$$R_{ij}(\theta) = \frac{\sqrt{\varepsilon_i}\cos\theta - \sqrt{\varepsilon_j - \varepsilon_i \sin^2\theta}}{\sqrt{\varepsilon_i}\cos\theta + \sqrt{\varepsilon_j - \varepsilon_i \sin^2\theta}},\qquad(4)$$

where $\varepsilon_i$ is the subsurface material permittivity. By analyzing the expression of the subsurface echo power presented in Eq. (3), we notice that there is a similarity between the atmospheric attenuation term in the bat case and the subsurface attenuation in the radar case. This represents a further relevant analogy that plays an important role in the definition of the bio-inspired ambiguities detection model described in the next section.

**Bio-inspired ambiguities detection.** Big brown bats adopt a combination of path attenuation difference and antenna pattern diversity[14] to distinguish foliage clutter. We can better understand the bat processing by analyzing the ratio of the echo power received at the two different harmonics (see Eq. (1)), which is equal to:

$$\Delta P_b(\theta, d) = \frac{P_{b,1}(\theta, d)}{P_{b,2}(\theta, d)} = \frac{G_{tr}(\theta, f_1)\, \sigma_b^0(\theta, f_1)}{G_{tr}(\theta, f_2)\, \sigma_b^0(\theta, f_2)}\, e^{-2(\alpha_b(f_1) - \alpha_b(f_2))d}.$$

$$(5)$$

This power ratio is only function of the frequency-dependent parameters such as attenuation and sonar cross-section of the target. This remarkable property implies that computing the echoes power ratio between different harmonics is a simple clutter discrimination strategy because it discards many physical parameters affecting the echoes intensity such as the power of the sonar call. It is interesting to note that the power ratio $\Delta P_b(\theta, d)$ defines a polar plot as function of the distance $d$ and the angle $\theta$. Except for the antenna gain, the other parameters such as attenuation are physical quantities not under the control of the bat. By analyzing the experimental results of Bates et al.[14], we can infer that echolocation evolution[20,21] shaped the bat's antenna gain at different harmonics to result in a functional and wise representation of the power ratio $\Delta P_b(\theta, d)$. Accordingly, it has very different values for nadir direction (i.e., target position) with respect to off-nadir direction. A potential ambiguous return can be identified by simply comparing the power ratios (Fig. 5a) of

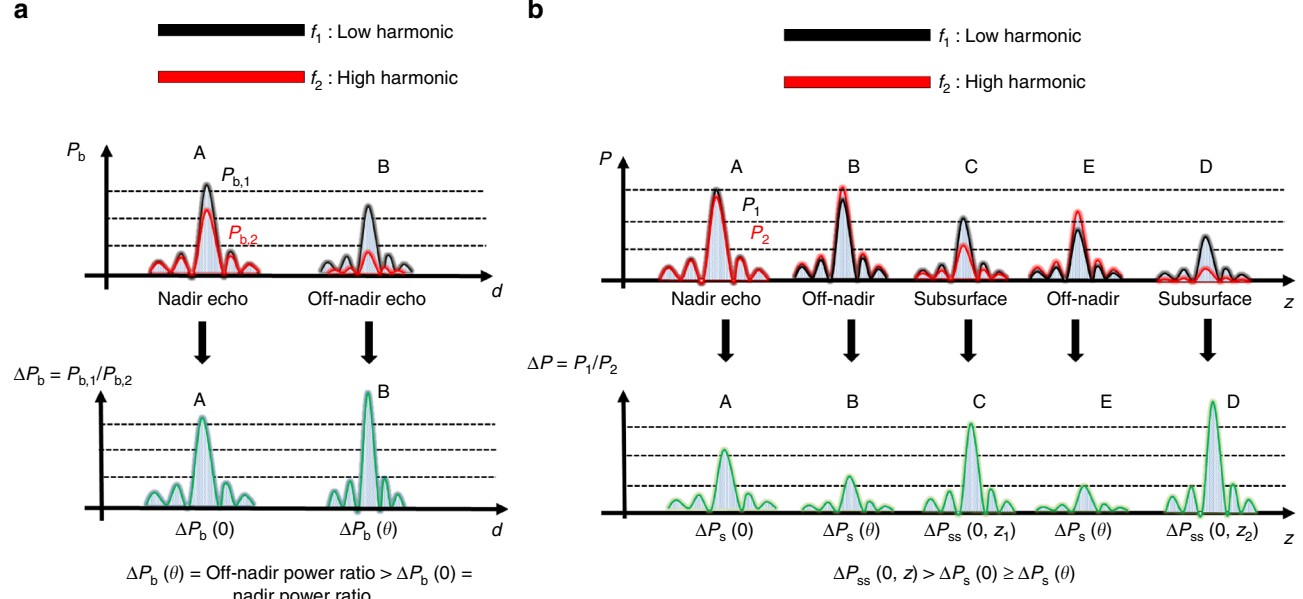

**Fig. 5** Bats and radar sounder disambiguation strategy. Visual examples on how bats and radar sounders are able to disambiguate clutter on the power return at different harmonics. **a** For the bat case, $P_{b,1}$ and $P_{b,2}$ are the echo power received as a function of the distance at the two different harmonics. Points A and B are referred to Fig. 4a. **b** For the radar sounder case, $P_1$ and $P_2$ are the echo power received as a function of the distance at two different harmonics. Points A to E are referred to Fig. 4b

the different received echoes and verifying if the power ratio of a potential echo coming from nadir direction is greater than the one of an echo originated from a different angular direction such that:

$$\Delta P_b(0, d_1) < \Delta P_b(\theta, d_2), \quad (6)$$

where $d_1$ represents the distance between the bat and the target and $d_2$ the distance between the bat and the ambiguous return (e.g., foliage). Please note that in this paper by convention, the power ratio definition is reversed if compared to the experimental results of Bates et al.[14].

In planetary radar sounders, the antenna type (i.e., dipole) excludes the possibility of having two antenna patterns with very different beam widths. According to this, we assume that $G^2(\theta, f_1) \simeq G^2(\theta, f_2)$. On the other hand, the radar sounder signal experiences a greater attenuation in the nadir direction with respect to a bat signal due to the presence of the subsurface. It is important to note that both antennas are pointed toward nadir direction. Following the bat clutter mitigation strategy, we define the surface echo power ratio $\Delta P_s(\theta)$ as follows:

$$\Delta P_s(\theta) = \frac{P_{s,1}(\theta)}{P_{s,2}(\theta)} = \frac{f_2^2 \, \sigma_s^0(\theta, f_1)}{f_1^2 \, \sigma_s^0(\theta, f_2)}. \quad (7)$$

The subsurface echo power ratio $P_{ss}(\theta, z)$ is equal to:

$$\Delta P_{ss}(\theta, z) = \frac{P_{ss,1}(\theta, z)}{P_{ss,2}(\theta, z)} = \frac{f_2^2 \, \sigma_{ss}^0(\theta, f_1)}{f_1^2 \, \sigma_{ss}^0(\theta, f_2)} \, e^{-2(f_1 - f_2)\alpha z}. \quad (8)$$

The result of Eq. (8) is similar to the surface echo power ratio of Eq. (7), but it has an additional dependence on the subsurface attenuation. The exponential scaling factor is always positive since we consider that $f_1 < f_2$.

Being the surface and subsurface natural terrains, we describe the backscattering coefficients assuming a fractal model[22,23]. We derive the surface and subsurface echo power ratio for the nadir-

looking case (i.e., $\theta = 0$), which are, respectively, equal to:

$$\Delta P_s(0) = \left(\frac{f_2}{f_1}\right)^{2/H_s} \quad (9)$$

$$\Delta P_{ss}(0, z) = \left(\frac{f_2}{f_1}\right)^{2/H_{ss}} e^{-2(f_1 - f_2)\alpha z}, \quad (10)$$

where $H_s$ and $H_{ss}$ are the Hurst exponents of the surface and subsurface, respectively. The value of the Hurst exponent is related to the terrain roughness. If the following condition is verified, then the subsurface echo signal can be always discriminated from clutter:

$$\Delta P_{ss}(0, z) > \Delta P_s(0) \geq \Delta P_s(\theta). \quad (11)$$

This inequality is inferred from the big brown bat clutter cancellation scheme previously described with the obvious addition of the subsurface return. An illustration describing how ambiguity can be resolved in the radar sounding case is shown in Fig. 5b. In order to apply the disambiguation condition, it is useful to locate the surface echo return. In general, this can be easily done for each echo trace as it is the return with the highest intensity[24]. The reader can notice that when compared to the bat case of Eq. (6), the condition is inverted. In the radar case, an higher attenuation difference among the two harmonics in the nadir direction is useful for improving clutter detection. In the bat case, the validity of the disambiguation condition of Eq. (6) has been experimentally verified. In the case of the radar sounder, to verify the limits of validity of the above disambiguation condition, first we need to analyze the inequality $\Delta P_{ss}(0, z) > \Delta P_s(0)$. This is equal to:

$$\frac{f_2 - f_1}{\ln(f_2/f_1)} > \frac{H_{ss} - H_s}{H_{ss} H_s} \frac{1}{\alpha z}. \quad (12)$$

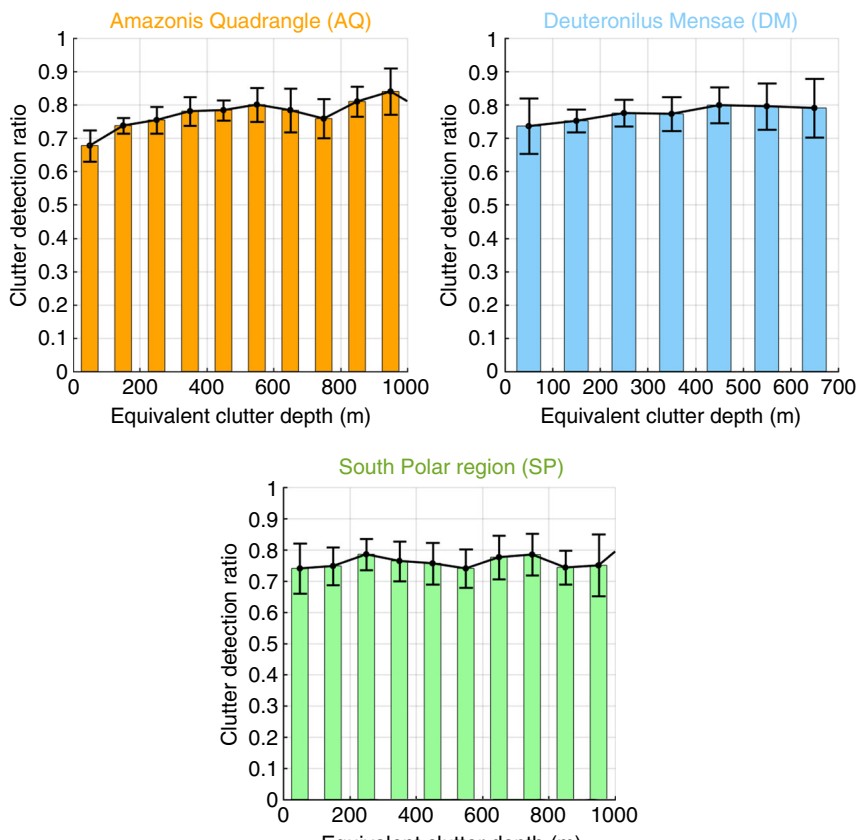

**Fig. 6** Clutter detection results. Clutter detection ratio that is defined as the number of clutter feature correctly classified as clutter over the total number of clutter features detected at each given depth, and related mean absolute deviation versus the equivalent penetration depth for the different data sets. The statistics were computed considering the SHARAD tracks displayed in Supplementary Fig. 4. The maximum equivalent clutter depth depends on the surface roughness. The rougher the surface the higher the measured maximum equivalent clutter depth for a given data set

Equation (12) allows the selection of radar sounder harmonics in terms of (i) surface/subsurface geometrical parameters (i.e., $H_s$ and $H_{ss}$), (ii) dielectric properties of the subsurface (i.e., $\alpha$), and (iii) penetration depth $z$. If we consider fixed the aforementioned listed parameters, then the model performance in detecting ambiguities increases as the two harmonics separation increases. This is because the difference in power ratio of the subsurface with respect to the surface is equal to:

$$\Delta P_{ss}(0,z)/\Delta P_s(0) = (f_2/f_1)^{2c_h} e^{2\alpha(f_2 - f_1)z}, \qquad (13)$$

where $c_h = (H_s - H_{ss})/H_{ss}H_s$. Equation (13) provides an indication of the model sensitivity.

The dielectric properties of the subsurface play an important role in ambiguity detection performance. The subsurface two-way attenuation factor can be written as $\alpha = 2\pi/c \tan \delta \sqrt{\varepsilon}$, where $\varepsilon$ is the real part of the dielectric constant of a given subsurface material and $\tan\delta$ the loss tangent[25]. High subsurface attenuation scenarios resulting in larger values of $\alpha$ provide better clutter disambiguation performance both in terms of sensitivity (i.e., $\Delta P_{ss}(0,z)/\Delta P_s(0)$) and minimum penetration depth (see Eq. (12)) when compared to low attenuation ones. The hypothesis that the surface echo power ratio from nadir direction is greater than the one from off-nadir direction (i.e., $\Delta P_s(0) \geq \Delta P_s(\theta)$) can be proven according to physical considerations. If we assume $f_2 > f_1$ in a confined range of frequencies typical of an orbital radar sounder, then it is always true that the power ratio $\Delta P_s(\theta)/\Delta P_s(0)$ will diminish for an increasing off-nadir angle $\theta$. This is because the lower harmonic $f_1$ always perceives a smoother surface compared

to $f_2$. This results in a stronger backscattering at $f_1$ for small off-nadir angles when compared to $f_2$. On the other hand, the backscattered power at $f_1$ decays faster than the return at $f_2$ as the off-nadir angle increases. Thus, the result is a reduction of the ratio $\Delta P_s(\theta)/\Delta P_s(0)$ as $\theta$ increases. This hypothesis is supported by considering the frequency-dependent root mean square slope $s$ $(f_2, f_1)$ of a given surface defined as follows[26]

$$s(f_2, f_1) = s_0 \left(\frac{f_2}{f_1}\right)^p \quad 0 \leq p \leq 1, f_2 > f_1, \qquad (14)$$

where the reference slope is denoted as $s_0$. It is clear from Eq. (14) that surface roughness increases as $f_2$ increases for a fixed $f_1$. Being the slope, an indicator of a given surface roughness, this proves our hypothesis that $\Delta P_s(0) \geq \Delta P_s(\theta)$. Large facets of sloping terrains are among the main contributors to clutter echoes, which could be typically mistaken as subsurface reflections. The value of $\theta$ of the surface and subsurface backscattering function is affected by the local slope of this type of features. This phenomenon could improve or degrade the disambiguation performance depending on the actual value of the feature surface slope and thus is strictly scenario dependent. In general, the value of $\theta$ in the nadir region should be smaller than the one in the off-nadir region. This is required to fully preserve the validity of the disambiguation condition on clutter and is

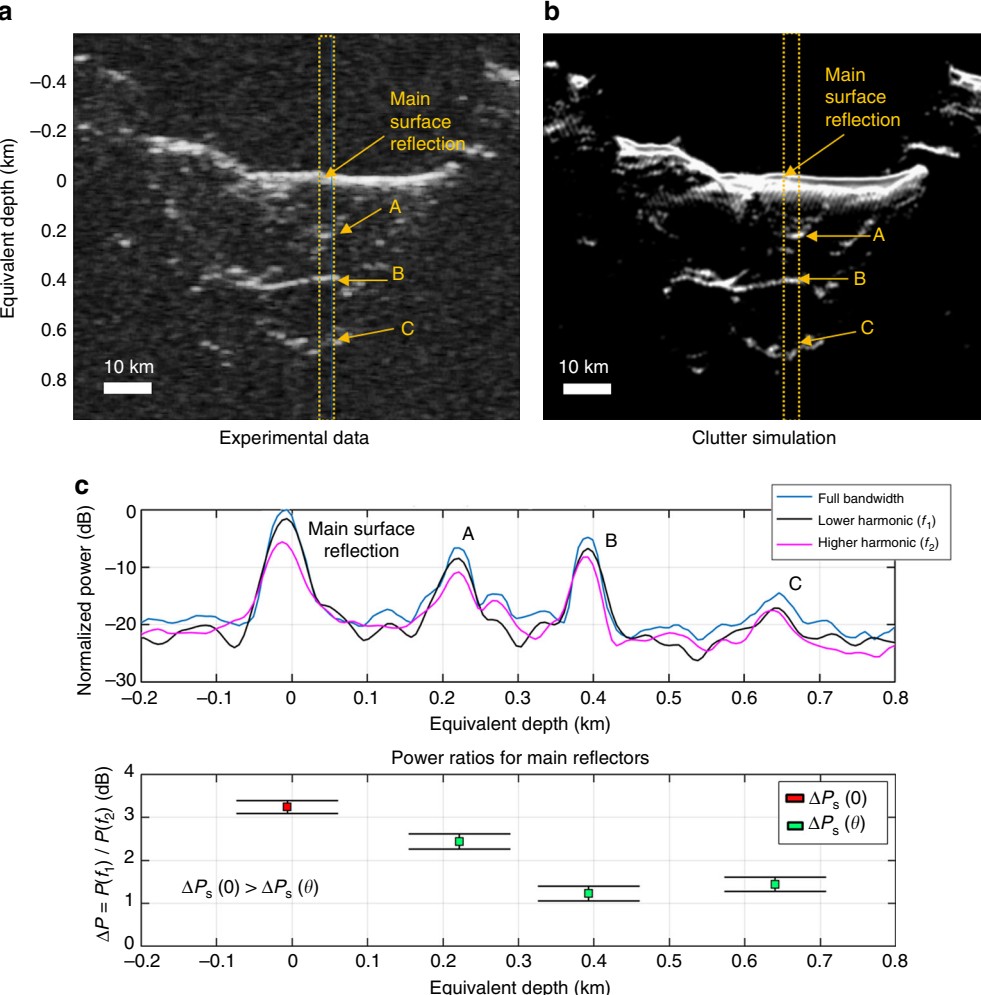

**Fig. 7** Example of clutter detection in the South Polar region. Example of the bio-inspired model results for a portion of the SHARAD radargram 0263001. The radargram is acquired over South Polar region near Promethei Rupes and contains only clutter features that resemble subsurface echoes. The disambiguation condition on clutter $\Delta P_s(0) \geq \Delta P_s(\theta)$ is verified allowing the clutter discrimination to be effectively performed. **a** SHARAD radargram 0263001. The dashed orange box is the region under investigation. Main off-nadir clutter reflector regions are marked with letters from A to C. **b** Clutter simulation confirming that the reflections are generated by surface features only. In this case, the clutter is generated by different craters walls. **c** Upper plot depicts echo traces (average over the investigated region) for the full signal and the harmonics decomposition. The lower plot shows the mean and standard deviation of power ratios for main reflectors. The number of echo traces considered for deriving the statistics is equal to 500. The equivalent depth in the subsurface medium has been computed from time delay assuming $\varepsilon = 3.1$

mathematically expressed as:

$$\underbrace{\cos^{-1}\left(\hat{d}_n \cdot \hat{n}_n\right)}_{\text{Nadir backscattering } \theta} \leq \underbrace{\cos^{-1}\left(\hat{d}_o \cdot \hat{n}_o\right)}_{\text{Off-nadir backscattering } \theta} , \quad (15)$$

where $\hat{d}_n$ and $\hat{d}_o$ are the distance versors pointing from any generic nadir and off-nadir surface locations to the radar, respectively (Supplementary Fig. 1). Similarly, we denote as $\hat{n}_n$ and $\hat{n}_o$ the local surface normal to any given nadir and off-nadir surface point, respectively, and as (·) the dot product. Indeed, the local value of the surface normal is directly connected to the value of the local surface slope. The experimental results presented in the following section show that the local surface slope does not have a major impact on the model performance and the condition of Eq. (15) is in the majority of cases verified.

**Performance evaluation.** We analyzed the effectiveness of the devised bio-inspired model to discriminate clutter features, which can be disguised as subsurface structures by using real radar

sounder data acquired over Mars. We verified the validity of the proposed model by analyzing radargrams under three different hypothesis: (a) clutter signal only, (b) subsurface signal only, and (c) mixed case, where both clutter and subsurface signals are present. In order to test the proposed model for different surface and subsurface conditions, we selected three regions of Mars namely the South Polar (SP), the Deuteronilus Mensae (DM), and the Amazonis Quadrangle (AQ). The SP and DM regions are icy regions[5,27]. The AQ is a volcanic region and its subsurface is mainly composed by dry sediments[28]. The subsurface-only and clutter-only hypotheses have been tested on all the data sets. The mixed case hypothesis has been validated on the AQ and DM data sets, as the SP data set did not provide sufficient statistical data for the mixed case. This is because in the SP region, the subsurface structures are in locations where the surface topography is relatively smooth and does not generate clutter.

The experiments for the different data sets were performed on the Shallow Radar (SHARAD)[29] reduced data records (RDR). SHARAD is currently orbiting and operating around Mars and its main parameters are listed in Supplementary Table 1. The

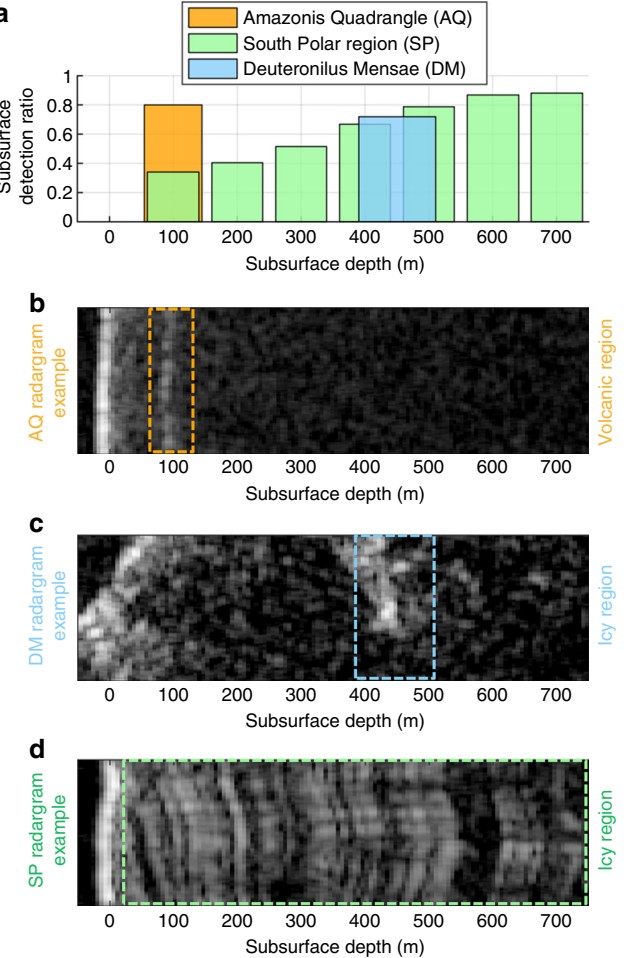

**Fig. 8** Experimental results on subsurface detection ratio (signal-only hypothesis) for the different data sets and AQ, DM, and SP regions radargrams examples. **a** Subsurface detection ratio, defined as the number of subsurface features correctly classified as subsurface versus the total number of detected features, at each given subsurface depth. The plot highlights the difference in subsurface detection ratio performance between a volcanic and an icy region due to the different subsurface attenuation values. **b** Typical radargram for the AQ region. The subsurface stratigraphy is such that the features are clustered around an average equivalent depths of 100 m. **c** Typical radargram for the DM region. The subsurface features are clustered around an average equivalent depth of 450 m. **d** Typical radargram for the SP region. The typical subsurface layering structure of this region allows the computation of statistics to be performed from 50 to 750 m depth

SHARAD RDR records consist of radar received echoes that have undergone basic radar processing. They are correlated with the auxiliary information needed to locate the observation in space and time and compensated for the ionosphere effect[30,31] resulting from the interaction between the radar signal and the martian ionosphere plasma[32]. Among the available data sets from planetary sounder instruments, the SHARAD ones are the only that allows to have both a reasonable harmonic separation and the same acquisition geometry, which is required by the proposed model.

Being SHARAD a single frequency channel system, the two harmonics and the associated bandwidths were retrieved by dividing the radar bandwidth into two non-overlapping sub-bands[33] (Supplementary Fig. 2). Accordingly, we obtained two linear frequency-modulated signals of bandwidth 5 MHz centered

at $f_1 = 17.5$ MHz and $f_2 = 22.5$ MHz, respectively. The data have been processed according to the procedure described in the "Methods" section for denoising and validation purposes.

For each analyzed radargram ambiguous and real subsurface returns were labeled for validation purposes by detecting them with a specific clutter echo simulator[34]. We defined an ambiguous return as a geological-like structure having a contiguous extent in both horizontal (i.e., along-track) and vertical (i.e., depth) direction. For each echo ambiguous return, the equivalent depth is estimated by considering the difference between the vertical value of its coordinates centroid and the estimated position of the surface echo (Supplementary Fig. 3). We limited the analysis to the first 1000 m of subsurface depth according to the nominal penetration capability of the SHARAD instrument. The specific SHARAD RDR used for validating the disambiguation condition are those listed in Supplementary Tables 2, 3, and 4.

In the clutter-only hypothesis, we verified whether the surface echo power ratio (i.e., $\Delta P_s(0)$) is greater or equal than the off-nadir surface power ratio (i.e., $\Delta P_s(\theta)$) according to the the proposed bio-inspired model. The ground tracks of the selected radargrams are shown in Supplementary Fig. 4. The results are analyzed in terms of the clutter detection ratio, which has been defined as the number of clutter feature correctly classified as clutter (i.e., $\Delta P_s(0) \geq \Delta P_s(\theta)$) over the total number of clutter features detected at each given depth. The experimental results for the three data sets (Fig. 6) show that the clutter detection ratio is satisfactory and nearly constant as a function of the clutter equivalent depth (i.e., the free space depth scaled by the expected value of the dielectric constant for the region under investigation). Moreover, the clutter detection ratio is similar for the different Mars regions under consideration. The SP and AQ regions exhibit deeper clutter features than the DM region (Fig. 6) due to the higher roughness of the surface as confirmed by the Mars laser altimetry data[35]. Interestingly, the values and the behavior of $\Delta P_s$ versus the depth are similar among the different investigated regions (Supplementary Fig. 5).

Figure 7 shows an example of results obtained in the SP region for the clutter-only hypothesis. The example illustrates how the disambiguation condition effectively discriminates the clutter generated by different craters, which can be mistaken as subsurface features. In this specific case, the value of $\Delta P_s(0)$ is equal to 3.1 dB. This value is in agreement with our model (Eq. (9)) for a surface having an Hurst exponent equal to $H_s = 0.7$, which corresponds to what is reported in the literature[36]. In the example, the values of $\Delta P_s(\theta)$ are provided as a function of depth. We analyzed the effects of the craters local slopes on the angle $\theta$ by computing the theoretical value of $\Delta P_s(\theta)$ and then comparing it with the experimental results shown in the example. The comparison (Supplementary Fig. 6) shows that there is good agreement between the model and the surface power ratio values, and that the off-nadir reflections surface slope result in a larger $\theta$ when compared to the nadir surface reflection.

In the case of subsurface signal only, we verified the validity of left-hand side of the disambiguation condition (i.e., $\Delta P_{ss}(0, z) > \Delta P_s(0)$). Similarly to the clutter-only case, we defined the detection ratio as the number of subsurface features correctly classified as subsurface versus the total number of detected features at each given depth. In the SP region, the subsurface stratigraphy allowed to compute the detection ratio from 50 to 750 m depth. Even though some subsurface structures are present from 750 to 1000 m, the SNR for both harmonics was too low for performing meaningful analysis (Supplementary Fig. 7). The experimental results show that the subsurface detection ratio increases as a function of the equivalent subsurface depth and reaches values larger than 0.8 for depths greater than 400 m (Fig. 8). The explanation for this is that, for small depths,

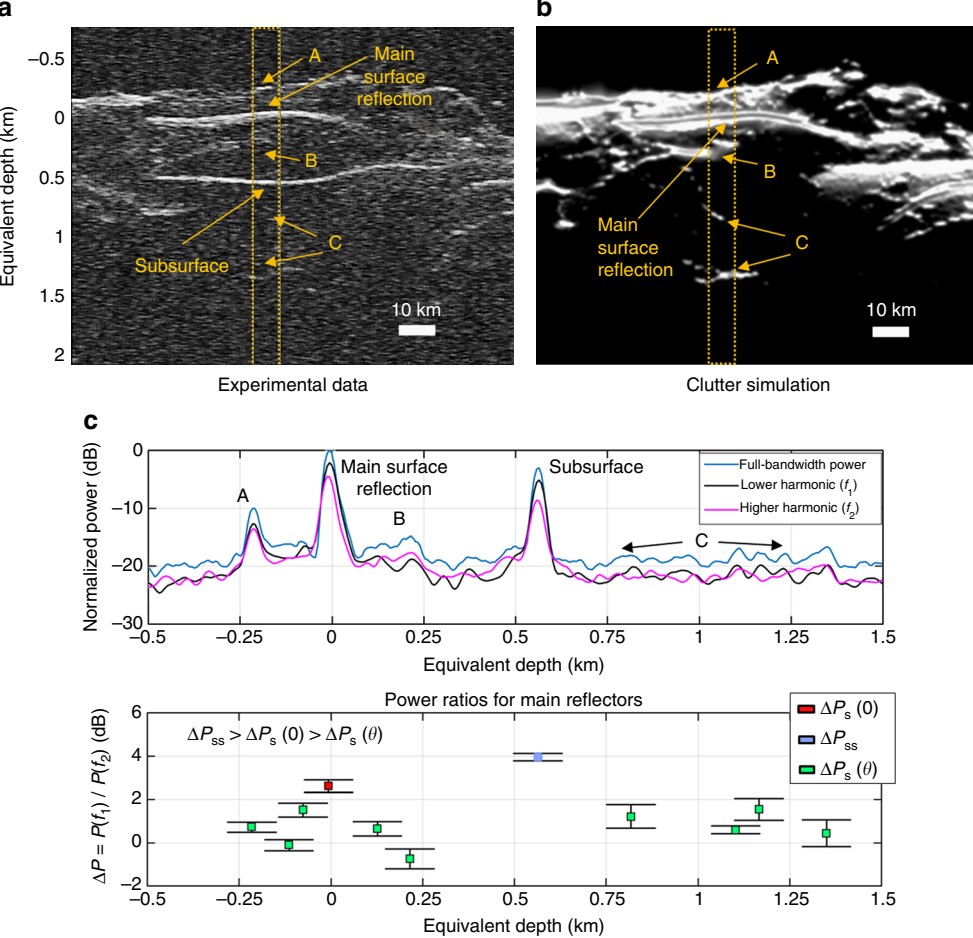

**Fig. 9** Example of clutter and signal detection in the Deuteronilus Mensae region. Example of the bio-inspired model results for a portion of the SHARAD radargram 0716902. The radargram is acquired over Deuteronilus Mensae and contains both clutter and subsurface features. The disambiguation condition $\Delta P_{ss}(0, z) > \Delta P_s(0) \geq \Delta P_s(\theta)$ is verified allowing the discrimination of clutter to be accurately performed. **a** SHARAD radargram 0716902. The dashed orange box is the region under investigation. Off-nadir clutter reflections are marked with letters from A to C. **b** Clutter simulation confirming that the reflections marked as A, B, and C are generated by surface features only and that in the experimental data there is a legitimate subsurface reflection. **c** Echo traces (average over the investigated region) for the full and split bandwidth SHARAD signal (upper plot) and mean and standard deviation of power ratios for main reflectors (lower plot). The number of echo traces considered for deriving the statistics is equal to 700. The equivalent depth in the subsurface medium is computed from time delay assuming $\varepsilon = 3.1$

the attenuation difference between $f_1$ and $f_2$ is not sufficient to effectively discriminate the subsurface signal thus resulting in a small detection ratio. As the depth increases, the attenuation difference increases resulting in an improved detection ratio. This is in line with what estimated by the proposed model (Eq. (13)), which predicts that for low depths the SP region represents a challenging case in terms of disambiguation performance since icy subsurface materials have low losses in the MHz range. For the DM data set, the measured average equivalent depth of the reflectors is equal to 450 m. The subsurface detection ratio is about 0.72. The average detection ratio for the AQ data set is of about 0.8 for subsurface features located at equivalent depths of 100 m. For the DM and AQ data sets, the clustering of subsurface features around a specific depth is in agreement with the geophysical analyses reported in the literature[5,28]. The comparative analysis of the subsurface detection ratios (Fig. 8) confirms that in SHARAD data the subsurface attenuation plays a major role in determining the performance of the method (Eq. (13)). The AQ data set provided solid disambiguation performance for low penetration depths when compared to the results of the SP region. This is expected being AQ a volcanic terrain with an higher two-way attenuation factor with respect to the other two

subsets. The DM data set results are in agreement with the ones of SP. This is also expected since the two regions share very similar subsurface material properties. The comparison between the theoretical values of $\Delta P_{ss}(z)/\Delta P_s(0)$ of Eq. (13) and the experimental values of the subsurface power ratios for the various data sets show that there is good agreement between the model and the expected geoelectrical properties of the subsurface (Supplementary Fig. 8). For the AQ data set, the model prediction is in agreement with the experimental data for values of loss tangent and dielectric constant consistent with those expected on a volcanic region[28]. Similar conclusions on model agreement can be done for the SP and the DM data sets using typical geoelectrical values for these icy regions[5,27].

In the mixed case hypothesis, we verified the validity of the combined disambiguation condition $\Delta P_{ss}(0, z) > \Delta P_s(0) \geq \Delta P_s(\theta)$. The experimental results show that the results obtained for the other two hypothesis are generalized by the mixed case. For the DM data set, the measured average power ratios are equal to $\Delta P_s(0) = 1.86 \pm 0.53$ dB, $\Delta P_{ss}(0, z \sim 100m) = 3.72 \pm 0.58$ dB and $\Delta P_s(\theta) = 0.59 \pm 0.55$ dB thus confirming the validity of the disambiguation condition. For the AQ data set, we measured average power ratios equal to $\Delta P_s(0) = 0.51 \pm 0.42$ dB, $\Delta P_{ss}(0, z \sim$

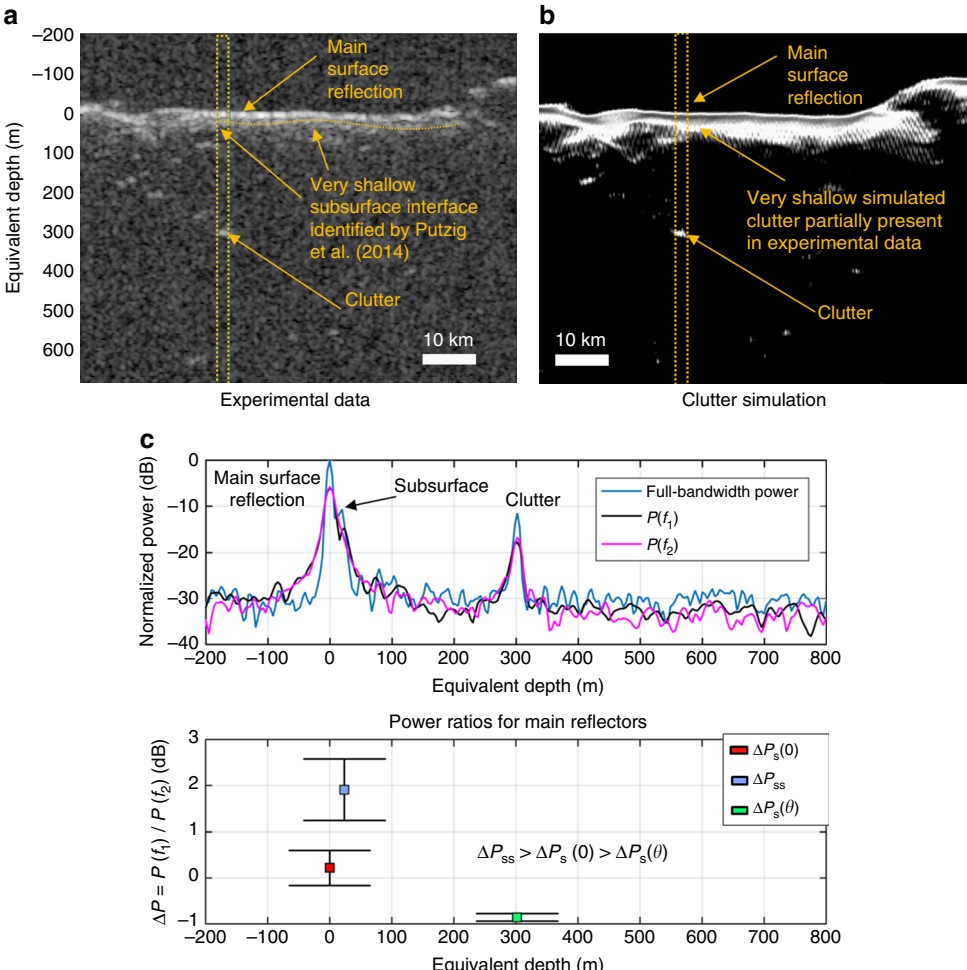

**Fig. 10** Example of clutter and signal detection at the Phoenix landing site in the Northern Plains of Mars. Example of the bio-inspired model results for a portion of the SHARAD radargram 1290301. This radargram has been previously analyzed by Putzig et al.[38] and it is acquired over the Green Valley of Vastitas Borealis and contains both clutter and subsurface features. The disambiguation condition $\Delta P_{ss}(0, z) > \Delta P_s(0) \geq \Delta P_s(\theta)$ is verified allowing the discrimination of clutter to be accurately performed and confirming the presence of a very shallow subsurface interface below the Phoenix landing site region. **a** SHARAD radargram 1290301. The dashed orange box is the region under investigation. The dashed orange line highlights a very shallow subsurface reflection mapped by Putzig et al.[38] using the same SHARAD observation. **b** In this case, clutter simulation highlights a clear clutter reflection at about 300 m depths but overestimates clutter in the shallow subsurface. **c** Echo traces (average over the investigated region) for the full and split bandwidth SHARAD signal (upper plot) and mean and standard deviation of power ratios for main reflectors (lower plot). The number of echo traces considered for deriving the statistics is equal to 400. The equivalent depth in the subsurface medium is computed from time delay assuming $\varepsilon = 8$

$450m) = 2.24 \pm 0.66$ dB and $\Delta P_s(\theta) = -0.34 \pm 0.27$ dB, once again confirming the disambiguation condition validity. Figure 9 shows an example of mixed case hypothesis result for the DM region. By applying the disambiguation condition, the subsurface feature located at about 550 m depth is correctly detected. The ambiguous reflections arising from different geological structures satisfy Eq. (11) and thus are correctly classified as clutter. The value of $\Delta P_s(0)$ is equal to 2.6 dB which, according to our model, corresponds to $H_s = 0.84$. Similarly to the previous discussed experiment, this value is in agreement with the literature. Another example of mixed case hypothesis result is provided in Fig. 10. This SHARAD radargram, acquired in the Phoenix[37] landing site region in the northern plains of Mars, has been already interpreted by Putzig et al.[38]. In their work, they highlighted the presence of a shallow subsurface layer (marked in Fig. 10a), which can correspond to a relatively deep base of ground ice ($\varepsilon = 3.15$) or to ice-free sediments (e.g., lava flows with $\varepsilon = 8$). The presence of the mapped subsurface layer is accurately detected by the proposed method (Fig. 10c) along with the discrimination of a clutter return located at about 300 m depth.

Despite the small harmonic ratio of the considered data (i.e., $f_2/f_1 = 1.29$) dictated by the lack of planetary radar sounder systems with larger harmonic ratios, the disambiguation condition provided satisfactory results for the different tested hypotheses, except for low depths in the SP data set. Obviously, a small harmonic separation makes the model more prone to detection errors due to additive random noise and terrain backscattered power variations. As stated, the proposed bio-inspired model predicts that the sensitivity and, in turns, the ambiguity solving accuracy increases by increasing the two harmonics separation. An example of the improvement in method sensitivity by varying the harmonic ratio $f_2/f_1$ is shown in Supplementary Figs. 9 and 10. The plots have been obtained considering typical Mars geoelectrical values[39,40] and assuming $f_1 = 17.5$ MHz and $f_2$ variable.

## Discussion

Recent studies revealed how big brown bats can effectively discriminate between a given prey and unwanted clutter coming from their sonar scene background. In this study, we proved that

this really powerful and relatively simple processing strategy can be mathematically modeled and adapted to radars for geophysical exploration of planetary bodies thus improving the scientific interpretation of the acquired data. This has been achieved by observing and modeling many interesting parallelisms between the two apparently distant fields of biology and applied electromagnetism for planetary exploration. As a result, this work opens up for a new way of dealing with ambiguous returns in planetary subsurface probing radars without the need of having an additional sensor paired with it to produce a 3D surface model for correct data interpretation. The proposed bio-inspired model has been tested on real Mars experimental data and provided high detection rate (except for low penetration depths in the South Polar region case) despite the small harmonic separation dictated by limited existing data sets. The measured values of surface and subsurface power ratios are in good agreement with the proposed theoretical model. As expected, the model provides better clutter detection performance for high attenuation rates of the subsurface medium. The experimental results confirm that the fractal assumption on the backscattering coefficient is effective in modeling complex scenarios, such as reflections from crater rims and hills sides, which are the main features contributing to subsurface clutter. Moreover, the experimental results show that the change in the local surface slope induced by this type of features is well within the model theoretical assumption for the validity of the disambiguation condition. Beside the clutter and subsurface disambiguation capability, which is the core result of this paper, the proposed general approach can be useful to characterize the roughness of a given surface and subsurface terrain. This can be achieved by relating it to the change in surface and subsurface echo power ratio versus wavelength.

As a final remark, we point out that the proposed clutter detection model should be used to design radar sounder systems defined to have a sufficient large harmonic separation for optimizing performances. In the bat case, the harmonic ratio is equal to $f_2/f_1 = 2$, thus sensibly higher than in our performed experiments. This could possibly explain the greater clutter detection accuracy reported for bat tests when compared to our results. The average clutter detection ratio in the bat case is 0.95, which is $0.19 \pm 0.01$ higher than in our case. Nevertheless, comparing our results with those obtained in laboratory test on bats is not trivial due to the absence of noise sources in the bat case, which could further explain the slight discrepancy in the obtained detection performance. Therefore, this works leaves the open question on whether bat clutter detection performance will change in a real natural environment.

## Methods

**Data processing**. Let us define as $R(x, y)$ the SHARAD full-bandwidth radargram. Each value of this bidimensional function represents an echo intensity. The Cartesian coordinate $x$ denotes depth while the $y$ coordinate denotes the position of the sensor along the ground track. As a result of dividing the signal bandwidth into two sub-bandwidths, we obtain the "low-frequency" radargram denoted as $R_L(x, y)$ and the "high-frequency" radargram denoted as $R_H(x, y)$ (see Supplementary Fig. 11). For each sub-bandwidth, we apply a 128 points moving average in the $y$ direction and a 5 points moving average in the $x$ direction. This is needed for increasing the signal-to-noise ratio. Then, we compute the statistical noise power for each echo trace. In radar sounding, noise samples are always available within a given range line leading or trailing the actual surface and subsurface signal. We exploit this property to compute the statistical mean of noise power for the lower sub-bandwidth namely $NPM_1(y)$. Then the threshold $T$ is computed as:

$$T(y) = K \cdot NPM_1(y), \qquad (16)$$

where $K$ is an arbitrary constant defining the threshold level, which is experimentally set to 1.7 for all the data set. The threshold is computed over the lower harmonic signal since it is expected to have a greater SNR especially for higher penetration depths with respects to the signal acquired at the higher harmonic. All the samples of each sub-bandwidth below the threshold are

discarded from the processing. Then, the power ratio $R_L(x, y)/R_H(x, y)$ is performed only on the peaks of each echo trace, which is averaged with its two neighboring samples in order to deal with sensor resolution uncertainty. The bidimensional power ratio is subsequently averaged with a $3 \times 3$ averaging filter. The surface return is located for each given $y$. Then, the disambiguation condition is applied to each echo trace for all the processed samples after the surface return. Each feature (e.g., clutter or subsurface) is represented by many echo samples allowing to compute mean and standard deviation of the power ratio.

**Software specifications**. The performance evaluation section (i.e., validation stage) and all the relevant described processing have been implemented in MATLAB environment.

**Data availability**. SHARAD RDR are publicly available on the NASA planetary data system geoscience node at the following address http://pds-geosciences.wustl.edu/missions/mro/sharad.htm.

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

## Acknowledgements

This work has been supported under a contract with the Italian Space Agency (Agenzia Spaziale Italiana—ASI).

## Author contributions

L.C. identified the bio-mimetic features, conceived the experiments, and conducted the experiments. L.C. and L.B. envisioned the model, analyzed the results, and wrote the manuscript.

## Additional information

**Competing interests:** The authors declare no competing financial interests.

