## [Peer Review File · Nature Communications]

Reviewers' comments:

Reviewer #1 (Remarks to the Author):

This is a very well thought out transposition of a sonar clutter rejection method discovered in echolocating bats to a radar sounder problem in geophysics. In essence, because the transmitted radar beam is too broad to exclude off-axis clutter returns, some alternative to narrow beaming has to be used to distinguish the desired on-axis subsurface structures from off-axis reflectors at similar distances. Bats transmit two harmonic frequency sweeps and exploit the fact that the higher frequency in the second harmonic is weaker than the first harmonic when it returns from the off-axis clutter, whereas the harmonics have equal strengths from the desired on-axis subsurface structure. Nice work! I do not see immediately how Figs. 6 and 7 select the desired reflections and reject the clutter, so I suggest that the authors retry this explanation so it is clear how the desired structures are separated from the clutter.

Reviewer #2 (Remarks to the Author):

Review of: Solving for Ambiguities in Radar Geophysical Exploration of Planetary Bodies by Mimicking Bats Echolocation

By L. Carrer and L. Bruzzone

This paper describes the introduction of an approach to discriminating surface clutter from subsurface reflectors in planetary radar sounder data returned by instruments such as SHARAD and MARSIS. This novel approach is based on an approximation of how bats process echolocation information to separate prey from foliage.

The bat-inspired processing strategy to discriminate clutter is very interesting. However, I was disappointed to find no examples of the putative novel approach applied to SHARAD or MARSIS radargrams to demonstrate that clutter from surface sources is clearly distinguished from subsurface echoes. Thus, the paper falls short of providing a proof of concept. Also, the approach, in spite of the inspiration taken from bats, is not particularly novel because it amounts to a split-chirp technique that has been previously applied to SHARAD data. I strongly recommend that authors revise the paper to include appropriate examples and analyses of their split-chirp clutter discrimination process directly applied to SHARAD radargrams. I would expect that these radargrams would show clutter features with consistent ratios (power or brightness) with time-delay and variable (diminishing power or brightness) with time-delay for legitimate subsurface reflectors.

In its present form, I cannot recommend the paper for publication in Nature Communications. With the suggested addition, and a clear description of how their approach is different from existing split-chirp methods, the paper may be suitable for publication.

Other comments and suggestions are provided below.

p. 2, par. 1, Authors should include a reference to Ono et al., Science, 2009 on the result of the lunar sounder on the Kaguya spacecraft.

p. 11, par. 2, The "Discussion" section reads like a summary. The statement "In this study, we proved that this really powerful and relatively simple processing strategy. . ." is not justified because it is not adequately demonstrated in the paper. As mentioned above, the developed processing scheme needs to be directly applied to SHARAD or MARSIS radargrams - illustrating its utility.

Figure 10. Is the unit for the y-axis of these plots correct - clutter detection rate? Do the authors mean "ratio" rather than "rate"?

Reviewer #3 (Remarks to the Author):

Recommendation:

My assessment is that this manuscript is not ready for publication and that significant additional derivation of more complex geometries and analysis to existing data needs to be completed with successful results before consideration. As a result, I suggest to reject it in the current form.

Paper Overview:

The manuscript proposes a method for discriminating between off-surface and subsurface signals in radar sounding echograms using a technique roughly based on similar biological methods used by bats. Surface clutter in airborne/satellite-based radar sounding measurements is due to "unwanted" off-nadir surface reflections occurring at the similar time-delays as subsurface reflection, since the geometry results in both having larger time delays than the surface return. The basic method relies on comparing the echo power ratio reflections at two distinct frequencies:

Power Ratio: $\Delta(P) = P(f1)/P(f2)$

It then uses propagation and scatterings models to define some basic inequalities that are used as test to discriminate between: nadir surface, off nadir surface, and nadir subsurface where:

$\Delta(P)\{\text{nadir-subsurface}\} > \Delta(P)\{\text{nadir-surface}\} \geq \Delta(P)\{\text{off-nadir-surface}\}$

The first inequality, $\Delta(P)\{\text{nadir subsurface}\} > \Delta(P)\{\text{nadir surface}\}$, is justified by the assumption that propagation and attenuation through the subsurface medium will decrease the reflected power at higher frequencies ($f2$) more than at lower frequencies ($f1$), and as a result the ratio of $P(f1)/P(f2)$ will increase for subsurface reflections. There are some additional dependencies on surface roughness as well – which are presented in equation 12 of the text.

The second inequality, $\Delta(P)\{\text{nadir-surface}\} \geq \Delta(P)\{\text{off-nadir-surface}\}$, is justified by considering the backscatter roll off with angle at two different frequencies. It is expected that the roll off will be steeper (backscatter will decrease faster) at lower frequencies, and the result is that $P(f1)/P(f2)$ will decrease for off-nadir surface reflections.

The paper then applies the method to data collected from SHARAD sounding radar on-board MRO.

Comments:

I. Provide more background information on clutter rejection and discrimination for sounding radar.

II. Although the connection to Bats is interesting, I don't think it is required or at least in the extent to which it is presented in the manuscript, (few pages in total). The idea of using the frequency diversity to investigate radar reflections, especially for GPR, is frequently applied for target (interface) characterization.

III. The application to planetary sounders is difficult due to the constraints on the antennas and resulting bandwidths typically limit the frequency separation that can be achieved between $f2$ and $f1$. The MARSIS instrument on Mars Express might have been a more appropriate choice since the relative frequency span between the radar sounding bands is much greater than SHARAD, although there are additional problems with ionosphere dispersion (that SHARAD experiences as well) that are not mentioned in the manuscript.

IV. The presented model is overly simplified and seems by only having a planar rough surface/subsurface with some lossy medium between.

V. Depending on the physical properties of the material, the attenuation may or may not increase with frequency to the extent that it is detectable with the achievable bandwidth constraints of the antenna.

VI. The clutter problem itself can be separated into two different types of scenarios of surface scattering. One where the clutter masks the subsurface reflection, and the second where the clutter is mistaken for a subsurface reflection.

(1) When the surface roughness is sufficient enough such that the surface reflection itself will decay over many range cells. For this case the argument that the surface seems smoother at the lower frequency f_1 may be justified. However, if this is the case when the clutter would simply be identified as the part of the surface response.

(2) When there are large facets of sloping terrain (hills or crater walls) that are normal to the instrument and appear as nadir reflections relative to the local slope. This is the clutter that could be typically mistaken as subsurface reflections and is of most concern for discrimination. However, it is not clear that these scenarios relate to the presented theory, as a hillside or crater wall may have different roughness characteristics.

VII. It is interesting to see the clutter detection rate over various regions to show consistent values. A bit more detail should be given to actual meaning of "clutter detection rate". I am making an assumption that if we test all the reflections that occur after the surface – this is the fraction of those that have a $\Delta(P)$ less than the surface $\Delta(P)$. It would be beneficial to provide values for the $\Delta(P)$ and identify some specific examples (with callouts in some reduced bandwidth f_1 and f_2 echograms).

VIII. There needs to be some results presented with subsurface reflections as well in areas where there is significant clutter that could be mistaken as subsurface. This is a missing critical result that in my view needs to be included in the manuscript.

Reviewer #1 (Remarks to the Author):

[RC-1-1]: *I do not see immediately how Figs. 6 and 7 select the desired reflections and reject the clutter, so I suggest that the authors retry this explanation so it is cleared how the desired structures are separated from the clutter.*

Author's Answer: Thanks for the remark, We reworked the explanation and improved the figure (see Fig. 5). We also added plots on experimental results which we think help to clarify how the whole model works (see Figure 7 and Figure 9 in the revised paper).

Reviewer #2 (Remarks to the Author):

[RC-2-1]: *I was disappointed to find no examples of the putative novel approach applied to SHARAD or MARSIS radargrams to demonstrate that clutter from surface sources is clearly distinguished from subsurface echoes. Thus, the paper falls short of providing a proof of concept.*

Author's Answer:

We added examples of the proposed approach directly applied to SHARAD radargrams in the paper (e.g. Fig. 7 and Fig. 9 of the revised paper). Moreover, we provided extensive new analysis covering three different types of hypothesis namely: (a) clutter only (b) subsurface signal only and (c) mixed case (clutter + subsurface). This allowed us to fully verify that the proposed model and in particular the proposed disambiguation condition is verified (i.e. subsurface power ratio > nadir surface power ratio \geq off-nadir surface power ratio). We performed a broad variety of experiments in order to guarantee statistical significance of our results. Apart from the examples and analyses provided in the main revised paper, the full list of experiments is provided in the Supplementary Tables. We also performed tests and analyses to verify if the measured surface and subsurface power ratio values, apart from verifying the disambiguation condition, are also in line with what our model equation predicts for the SHARAD operating scenario. These analyses are reported as well in the main paper.

[RC-2-2]: *The approach, in spite of the inspiration taken from bats, is not particularly novel because it amounts to a split-chirp technique that has been previously applied to SHARAD data. I strongly recommend that authors revise the paper to include appropriate examples and analyses of their split-chirp clutter discrimination process directly applied to SHARAD radargrams. I would expect that these radargrams would show clutter features with consistent ratios (power or brightness) with time-delay and variable (diminishing power or brightness) with time-delay for legitimate subsurface reflectors.*

Author's Answer: Thanks for the remark. We added examples showing the proposed model directly applied to SHARAD radargrams (see Figure 7 and Figure 9 of the revised paper). We included extensive experiments for the different regions under investigation also for the subsurface and clutter plus subsurface (i.e. mixed) case. Indeed, the experimental results show that the clutter features have consistent power ratio slightly decaying with equivalent clutter off-nadir angle as predicted by the model due to change in backscattering angle. For the legitimate subsurface features, the power ratio is increasing with equivalent depth and difference in subsurface attenuation between volcanic and icy regions. Please note that we defined the power ratio as power of the low frequency channel over the power of the high frequency channel and not vice versa. Therefore the subsurface power ratio increases with depth and not diminishes.

For what regards the novelty of the proposed approach, we applied split-chirp to SHARAD data because radar sounder data with very different central frequencies are not available. As stated in the paper and proved mathematically, increasing the separation between the two central frequencies would be very beneficial for the proposed approach. Beside the clutter and subsurface disambiguation capability, which is the novel core result of this paper, the proposed general approach can be used in two specific sub-cases (i) to characterize the roughness (e.g. Hurst exponent estimate) of a given surface and subsurface and (ii) to estimate the subsurface geoelectrical parameters which is the sub-case the reviewer is referring to that has already been applied to SHARAD data. For the latter one, tight requirements on the value of the SNR for both harmonics shall be respected.

To give an example on the difference between our proposed approach and the split-chirp for subsurface parameters estimation which is a sub-case of our proposed approach, let us assume we sound the subsurface of Mars with two very different central frequencies namely $f_1 = 1$ MHz and $f_2 = 100$ MHz. For a vast range of subsurface materials and considering typical radar sounder parameters, it is reasonable to expect that f_1 will effectively propagate into the subsurface to much greater depths when compared to the f_2 signal. In this case, the proposed bio-inspired model predicts good performance for the disambiguation of clutter and subsurface signal because the attenuation difference between f_1 and f_2 is very large and the surface response will be different too as function of the wavelength. On the contrary, the tangent loss and dielectric constant estimation method will probably fail because the attenuation at f_2 is such that it will propagate very little in the subsurface when compared to f_1 , therefore making the inversion procedure, due to signal to noise ratio issues on f_2 signal, really difficult to be performed. In this case, the main claim of the proposed approach (i.e. clutter disambiguation based on two harmonics) based on the bio-inspired proposed model will continue to hold. The derived result (i.e. subsurface properties estimation) which is not the main focus of this paper and thus we consider it as a possible and indeed interesting derived result cannot be performed.

[RC-2-3]: *Authors should include a reference to Ono et al., Science, 2009 on the result of the lunar sounder on the Kaguya spacecraft.*

Author's Answer: Thank you, this is now included in the paper.

[RC-2-4]: *p. 11, par. 2, The "Discussion" section reads like a summary. The statement "In this study, we proved that this really powerful and relatively simple processing strategy. . ." is not justified because it is not adequately demonstrated in the paper. As mentioned above, the developed processing scheme needs to be directly applied to SHARAD or MARSIS radargrams - illustrating its utility.*

Author's Answer: Thanks for the remark. We included direct applications of the processing scheme to actual SHARAD data (e.g. Fig.7 and Fig.9 of the paper). We added extensive experiments for the different datasets considering a variety of test cases and adding discussions regarding the produced results in terms of agreement with our proposed model. For extensive details see answer [RC-2-2]. Discussion section has been improved too with additional comments of the obtained results, the main novelty of the method and possible additional applications.

[RC-2-5]: *Figure 10. Is the unit for the y-axis of these plots correct – clutter detection rate? Do the authors mean "ratio" rather than "rate"?*

Author's Answer: Thanks for the remark. We corrected to ratio. We also added a clarification in the experimental results section on the meaning of 'Detection Ratio'.

Reviewer #3 (Remarks to the Author):

[RC-3-1]: *Provide more background information on clutter rejection and discrimination for sounding radar.*

Author's Answer: We added a more detailed discussion and references on state-of-the-art on clutter rejection and discrimination for planetary radar sounding in the introduction section of the revised paper.

[RC-3-2]: *Although the connection to Bats is interesting, I don't think it is required or at least in the extent to which it is presented in the manuscript, (few pages in total). The idea of using the frequency diversity to investigate radar reflections, especially for GPR, is frequently applied for target (interface) characterization.*

Author's Answer: We prefer to keep the bats connection in the paper. In fact, the idea of exploiting frequency diversity for clutter detection has been derived by reading and analyzing biology publications on bats. Then we formulated a theoretical model. We did not discover the connection and analogy in reverse and empirical way (i.e. frequency diversity for radar to bats). Bats use their multi-frequency sonar for both clutter detection and targets (i.e. insects) characterization [1]. This is expected since their sonar system is mainly used for foraging and navigation. The bats clutter detection approach has been extensively discussed and verified in the literature [2]. We based our work on those scientific findings. How the bats perform target characterization with their sonar system is still debated in the literature [3]. According to this, the proposed bio-inspired approach offers a general model where the clutter detection is the core result which we presented in the paper. The interface characterization is just a derived results of our general model based on the bio-inspired approach (see also [RC-2-2]).

REFERENCES FOR ANSWER [RC-3-2]

[1] Nachtigall, Paul E., and Patrick WB Moore, eds. Animal sonar: Processes and performance. Vol. 156. Springer Science & Business Media, 2012.

[2] Bates, Mary E., James A. Simmons, and Tengiz V. Zorikov. "Bats use echo harmonic structure to distinguish their targets from background clutter." *Science* 333.6042 (2011): 627-630.

[3] Simmons, James A., and Lynda Chen. "The acoustic basis for target discrimination by FM echolocating bats." *The Journal of the Acoustical Society of America* 86.4 (1989): 1333-1350.

[RC-3-3]: *The application to planetary sounders is difficult due to the constraints on the antennas and resulting bandwidths typically limit the frequency separation that can be achieved between f_2 and f_1 . The MARSIS instrument on Mars Express might have been a more appropriate choice since the relative frequency span between the radar sounding bands is much greater than SHARAD, although there are additional problems with ionosphere dispersion (that SHARAD experiences as well) that are not mentioned in the manuscript.*

Author's Answer:

For what regards the applicability of the proposed model to planetary radar sounder (see also [RC-3-5]), in this paper we applied split chirp technique to SHARAD data due to the lack of existing sensors with larger frequency separation. An higher frequency separation is very likely to improve the performance of the model. As an example, the upcoming radar sounder REASON [1], which will sound Jupiter's moon Europa, is a multi-frequency radar with planned central frequencies of $f_1 = 9$ MHz and $f_2 = 63$ MHz. This frequency separation would be very suitable to use the proposed approach for clutter reduction.

According to Table [RC-3-3-1], MARSIS has four subsurface sounding modes with respect to probing central frequency. In detail, 1.3-2.3 MHz (centered at 1.8 MHz), 2.5-3.5 MHz (centered at 3.0 MHz), 3.5-4.5 MHz (centered at 4.0 MHz) and 4.5-5.5 MHz (centered at 5.0 MHz). The maximum carrier frequency difference useful to test the proposed bio-inspired model is achieved by selecting a MARSIS trace operating with 5 MHz centre frequency and one with 1.8 MHz centre frequency. In this case, the maximum achievable frequency difference is equal to $5.5 \text{ MHz} - 1.3 \text{ MHz} = 4.2 \text{ MHz}$ which is lower than the SHARAD case of $25 \text{ MHz} - 15 \text{ MHz} = 10 \text{ MHz}$. This computation does not take into account the bandwidth. In the SHARAD case the best splitting option is to consider 15-20 MHz (centered at

17.5 MHz) and 20-25 (centered at 22.5 MHz). In the MARSIS case the best possible option is to use 1.3-2.3 MHz (centered at 1.8 MHz) and 4.5-5.5 MHz (centered at 5 MHz) therefore having smaller central frequency difference and a very large difference in the bandwidth size greatly impacting resolution with respect to SHARAD (i.e. 1 MHz vs 5 MHz). As a final remark in favor of the usage of SHARAD data for our purpose, it is much less affected by ionosphere than MARSIS data as the reviewer remarked. We added to the paper a discussion regarding the ionosphere along with the relevant related literature in the experimental section.

REFERENCES FOR ANSWER [RC-3-3]

[1] Moussessian, A., et al. "REASON for Europa." AGU Fall Meeting Abstracts. 2015.

MARSIS subsurface sounding mode characteristics				
Centre frequency (MHz)	1.8	3.0	4.0	5.0
Bandwidth (MHz)	1.0	1.0	1.0	1.0
Source European Agency Official Website: http://sci.esa.int/mars-express/34826-design/?fbodylongid=1601 See also: Picardi, G., et al. "Performance and surface scattering models for the Mars Advanced Radar for Subsurface and Ionosphere Sounding (MARSIS)." Planetary and Space Science 52.1 (2004): 149-156.				

Table [RC-3-3-1]: MARSIS subsurface sounding mode characteristics

[RC-3-4]: *The presented model is overly simplified and seems by only having a planar rough surface/subsurface with some lossy medium between.*

Author's Answer: The inclusion of the fractal backscattering function in the model allow to derive the proposed model performance also for complex scenarios by controlling the value of the Hurst exponent. Please refer to answer [RC-3-6] for an in-depth explanation which also includes an in-depth analysis regarding the effect of the local slope.

[RC-3-5]: *Depending on the physical properties of the material, the attenuation may or may not increase with frequency to the extent that it is detectable with the achievable bandwidth constraints of the antenna.*

Author's Answer: Thanks for the remark. Indeed, the proposed model needs a minimum delta attenuation between the two harmonics to be effective. We investigated the effect of the physical properties of the subsurface medium on the proposed model in the experimental section. To give an example, the results on Amazonis Quadrangle region are better for lower depths when compared to the south polar region. This is due to the higher tangent loss and dielectric constant of the Amazonis regions (mainly volcanic) when compared to south polar Region (icy region). We also added more details about the contribution of the subsurface material properties and how they affect the proposed model in the section on bio-inspired ambiguities detection (in particular see marked changes in bold after equation (13)). It is important to stress that we used SHARAD data because, among all available radar sounder data set, is the one that allows the highest frequency separation (please see also answer [RC-3-3] which relates to this one). An higher frequency separation is very likely to improve the performance of the model. In fact, the upcoming radar sounder names REASON [1] which will sound Jupiter's Europa moon is a multi-frequency radar with planned central frequencies of 9 MHz and 63 MHz. This frequency separation would be very suitable for the proposed approach. Nevertheless, as explained in our paper, the model provided good results even with a relatively small separation provided by the SHARAD split-bandwidth approach.

REFERENCES FOR ANSWER [RC-3-5]

[1] Moussessian, A., et al. "REASON for Europa." AGU Fall Meeting Abstracts. 2015.

[RC-3-6]: *The clutter problem itself can be separated into two different types of scenarios of surface scattering. One where the clutter masks the subsurface reflection, and the second where the clutter is mistaken for a subsurface reflection.*

(1) When the surface roughness is sufficient enough such that the surface reflection itself will decay over many range cells. For this case the argument that the surface seems smoother at the lower frequency f_l may be justified. However, if this is the case when the clutter would simply be identified as the part of the surface response.

(2) When there are large facets of sloping terrain (hills or crater walls) that are normal to the instrument and appear as nadir reflections relative to the local slope. This is the clutter that could be typically mistaken as subsurface reflections and is of most concern for discrimination. However, it is not clear that these scenarios relate to the presented theory, as a hillside or crater wall may have different roughness characteristics.

Author's Answer to [RC-3-6]:

We thank the reviewer for the very interesting remark. We added a theoretical discussion on the local surface slope, which reflects the analysis provided in this answer, in the bio-inspired ambiguities detection section of the paper (see equation (15) and related text of the revised paper)). Analysis on the actual impact of local surface slope are also included in the experimental results and discussion sections of the paper. In the discussion section we also included considerations regarding the capability of the proposed model in describing complex geometries through the fractal assumption.

We can divide our answer into two parts; (a) analysis on how the Hurst exponent of the fractal model, which directly relates to the fractal backscattering function (i.e. sigma-zero) included in our model, it is able to describe complex geometries (e.g. craters and hills) and a second part denotes as (b) where the constraints on local backscattering angle (i.e. θ of $\sigma^0(\theta)$) given by the combination of planetary sounders S/C height and terrain local slope are discussed. This discussion complements the experimental results of the method obtained over craters and hills (see model result examples of Fig. 7 and Fig. 9 of the revised paper) shown in the paper addressing other comments by reviewers.

(a) The proposed model is based on fractal terrain assumption. In particular, we consider the fractal fractional Brownian motion (fBm) model, which is well established in the literature. This is reflected in the fractal backscattering function $\sigma^0(\theta, f, H, T)$ where θ is the backscattering angle (depends on the local slope), f the carrier frequency, $0 \leq H \leq 1$ the Hurst coefficient and T is the Topothesis [1-2]. In particular, T and H are the fractal parameters describing the surface characteristics. The Hurst exponent H is directly connected to the fractal dimension of the surface. It can be interpreted as a measure of the surface correlation and thus it is directly connected to roughness/topography. For what pertains the experimental results part and thus the fractal model applicability in the SHARAD case, values of the Hurst exponent H have been derived for the entire Martian surface [3]. An interesting property of the fractal backscattering function model is that for $H=1$ the function $\sigma^0(\theta, f, H, T)$ collapses into the one of a regular non-fractal surface and for $H=0.5$ is equal to the Hagfor's backscattering function [1-2]. According to this, the fractal backscattering model is able to capture a variety of different surface conditions. The surface is smoother for $H=1$ and increasingly rougher as H decrease toward 0.5. The backscattering model connects the Hurst exponent, and in turns the topography and local surface slope, to the backscattering angle θ . To prove that complex scenarios (e.g. hills / craters walls) are covered by the proposed model, we generated different fBm surfaces¹ [4] for various Hurst exponents and compute the backscattering angle (which is indeed local slope dependent). The radar is assumed to be positioned at the center of the scene and at a height of 255 km (similarly to SHARAD). From figures [RC-3-6-1]- [RC-3-6-3], it is clear that the proposed model dependence on Hurst exponent is able to model complex geometry such as crater or hills, which are commonly found on planetary surfaces. Moreover, the backscattering angle map shown in the figures has been computed to support the discussion of point (b) on local slope.

(b) We agree with the reviewer that a local slope can result in having an angle θ from off-nadir which could resemble nadir. It is also true that a local slope can actually increase the backscattering angle thus improving

¹ The code used for generating the 2D fBm surfaces is available at <https://it.mathworks.com/matlabcentral/fileexchange/38945-fractional-brownian-field-or-surface-generator>

the performance of the model (see Fig. [RC-3-6-4]). If we analyze the distribution of the backscattering angle for the SHARAD radargrams we used for the performance evaluation, we find that the nadir backscattering angle is very often close to 0 and hills and crater have a local backscattering angle much greater than 0 which is required by the model. This is because we selected radargrams with a reasonable surface and subsurface SNR otherwise it is meaningless to perform any type of analysis. A reasonable SNR of the main surface echo directly implies a reasonable roughness and slope at the scale of the radar wavelength in the nadir region (see the examples of Figure [RC-3-6-5] and [RC-3-6-6]). The data reported in the figures are from actual radargrams we used in our analysis. The backscattering angle can be decomposed as $\theta(x, y) = \theta_f(x, y) + \theta_t(x, y)$ where $\theta_f(x, y)$ is the backscattering angle of a flat surface and $\theta_t(x, y)$ is the contribution of the topography/local slope. The coordinates x, y are the generic cartesian coordinates of ground points. The nadir point is assumed to be located at $(x, y) = (0, 0)$. The value of $\theta_f(x, y)$ radially increases from nadir as shown in Fig. [RC-3-6-2]-[RC-3-6-3] (Synthetic results) and Fig. [RC-3-6-5]-[RC-3-6-6] (Experimental Results). Naturally, the values of $\theta_t(x, y)$ depends on the terrain type. From the theoretical point of view, the model disambiguation condition requires that the backscattering angle in the nadir region has to be smaller than the one in the off-nadir areas illuminated by the radar. In the revised version of the paper this condition, which takes into account the local surface slope and the radar acquisition geometry, has been summarized by using concise vector notation (see equation (15) and related text).

As already stated, we experimentally found that this condition is very often satisfied in the SHARAD experimental data used for model validation. This means that the local surface slope $\theta_t(x, y)$ in the nadir region is small (fair roughness condition) and this is directly connected to the discussion about surface SNR. Fig. [RC-3-6-5]-[RC-3-6-6] shows an example of the values of the backscattering angle for real experimental data, the provided backscattering angle histogram confirms our hypothesis regarding small nadir sloping.

We can infer that the model disambiguation condition on surface clutter will fail under a very rough surface assumption. Under this condition, the radar echoes from both surface and putative subsurface are so weak to the extent that they are barely visible or invisible on the radargram (very low SNR). A striking example of this behavior are the radargrams acquired over the Lycus Sulci region ($28.14^\circ\text{N } 215.53^\circ\text{E}$) near Olympus Mons. It is well known that the roughness induced by the furrows and ridges populating that region at SHARAD scale is such that echoes from the surface are barely detected by SHARAD. An example of this behavior is provided in Figure [RC-3-6-7] for a radargram acquired over the discussed region. The examples also includes a region where the nadir fair roughness condition applies resulting in a solid surface SNR and validity of our proposed model. An interesting fact is that the Hurst exponent for the example of Figure [RC-3-6-7] is about 0.6 for the very rough area and the backscattering angle map resembles the results of the simulation of Figure [RC-3-6-1]. The Deuteronilus Mensae and Amazonis Planitia examples (see Fig. [RC-3-6-5] and [RC-3-6-6]) Hurst exponent is in the range 0.7 to 0.8 and the backscattering angle map resemble the synthetic results of Figure [RC-3-6-2]. This is an additional proof that our model and in particular the adopted backscattering function based on fractal model is fair in representing complex geometries such as craters and hills. In general the Hurst exponent for Mars with high probability is in the range between 0.7 and 1 [3].

Figure [RC-3-6-1]: Generated Synthetic DEM and corresponding local backscattering angle θ in degrees for $H = 0.6$.

Figure [RC-3-6-2]: Generated Synthetic DEM and corresponding local backscattering angle θ in degrees for $H=0.8$.

Figure [RC-3-6-3]: (left) Generated DEM and (right) corresponding local backscattering angle θ for $H=0.95$. The surface is almost flat thus the local backscattering angle θ in degrees tends to be equal to the flat surface contribution which can be computed as $\theta_f = \tan^{-1}\left(\frac{\sqrt{x^2+y^2}}{h}\right)$ where h is the S/C height equal to 255 Km.

Figure [RC-3-6-4]: Upper figure illustrates how the backscattering angle changes with local surface slope. The change in backscattering angle can be favourable for the method or unfavourable depending on the direction of the normal of the local slope with respect to the radar slant range vector.

Figure [RC-3-6-5]: Left: MOLA Digital Elevation Model (Elevation in meters) for an example in Deuteronilus Mensae Regions (radargram id: r_1232403_001_ss19_700_a), Right: Local backscattering angle θ in degrees derived from the MOLA data, Bottom left: Histogram of backscattering angle θ retrieved from the backscattering angle map of the right figure, Bottom right: Ideal backscattering angle of a flat surface with highlighted the area dimension of the SHARAD footprint.

Figure [RC-3-6-6]: Left: Mola Digital Elevation Model (Elevation in meters), Right: Local backscattering angle θ in degrees derived from MOLA data and bottom: Histogram of local backscattering angle θ for an example in Amazonis Planitia Regions (radargram id: r_0921303_001_ss19_700_a).

SHARAD Radargram 0209801, Power [dB]

(a)

(b)

(c)

(d)

(e)

(f)

Figure [RC-3-6-7]: (a) SHARAD Radargram `r_0209801_001_ss19_700_a` (Lycurus Sulci Region), the surface SNR is expressed in dB (b) Cluttergram for the radargram under investigation (c) Local back scattering angle θ in degrees for very rough area (d) Local back scattering angle θ in degrees for fair roughness area, and (e) MOLA DEM for very rough area (f) MOLA Digital Elevation Model for fair nadir roughness area

REFERENCES FOR ANSWER [RC-3-6]

- [1] Franceschetti, Giorgio, et al. "Scattering from natural rough surfaces modeled by fractional Brownian motion two-dimensional processes." *IEEE transactions on antennas and propagation* 47.9 (1999): 1405-1415.
- [2] Bruzzone, Lorenzo, et al. "Subsurface radar sounding of the Jovian moon Ganymede." *Proceedings of the IEEE* 99.5 (2011): 837-857.
- [3] Orosei, R., et al. "Self-affine behavior of Martian topography at kilometer scale from Mars Orbiter Laser Altimeter data." *Journal of Geophysical Research: Planets* 108.E4 (2003).
- [4] Kroese, D. P., & Botev, Z. I. (2015). Spatial Process Simulation. In *Stochastic Geometry, Spatial Statistics and Random Fields*(pp. 369-404) Springer International Publishing, DOI: 10.1007/978-3-319-10064-7_12

[RC-3-7]: *It is interesting to see the clutter detection rate over various regions to show consistent values. A bit more detail should be given to actual meaning of "clutter detection rate". I am making an assumption that if we test all the reflections that occur after the surface – this is the fraction of those that have a $\delta(P)$ less than the surface $\delta(P)$. It would be beneficial to provide values for the $\delta(P)$ and identify some specific examples (with callouts in some reduced bandwidth f1 and f2 echograms).*

Author's Answer: The Reviewer interpretation is correct. Values of $\delta(P)$ are now provided for the datasets under consideration for clutter signal only, subsurface signal only and mixed case (subsurface+clutter) hypothesis. The actual values of the surface power ratio are discussed in the text and provided in Supplementary Figure S6 for the different datasets. Explained examples with figures showing the values of the power ratio for the different cases are now included in the revised paper (see Fig.7 and Fig. 9). We provided the values of the subsurface power ratio as well. We used the surface and subsurface power ratio to perform analysis in order to verify the proposed model agreement versus the experimental power ratios. Clarifications on the meaning of detection ratio have been added to the paper.

[RC-3-8]: *There needs to be some results presented with subsurface reflections as well in areas where there is significant clutter that could be mistaken as subsurface. This is a missing critical result that in my view needs to be included in the manuscript.*

Author's Answer: Extensive analysis has been included in the paper for the different cases. We reported (a) clutter signal only, (b) subsurface signal only and (c) clutter+signal cases. We provided examples of the proposed model results (see Fig 7 and Fig. 9) directly applied to SHARAD radargrams showing the values of the power ratios. We thoroughly investigated the proposed clutter disambiguation condition for different datasets under the different hypothesis and the results are in line with what the model predicts.

Reviewers' comments:

Reviewer #1 (Remarks to the Author):

Good work. The revision satisfies all of my concerns.

Reviewer #2 (Remarks to the Author):

Review of: Solving for Ambiguities in Radar Geophysical Exploration of Planetary Bodies by Mimicking Bats Echolocation (Revised)

By L. Carrer and L. Bruzzone

The revised paper while greatly improved does not address key concerns raised with the original paper. Specifically, it is still not made clear in the revision how the approach taken is not equivalent to a split-chirp technique that has been previously applied to SHARAD data. The addition of fig. 7 and 9 while helpful in illustrating clutter detection are not very convincing as to the general applicability of the technique. In the example of clutter features for the impact craters shown in fig. 7, there is no subsurface reflector clearly distinguished from the surface clutter features. The SHARAD radargram shown in fig. 9a is confusing. Clutter features appear above the surface return in time-delay or depth. How is this possible? An example is needed, preferably a published SHARAD radargram that has intermingled subsurface reflectors and surface clutter features, where clutter features can be unambiguously attributed to surface features via ground-range projection of the radargram onto the surface or using a clutter model derived from topography.

Although greatly improved, I still cannot recommend the paper for publication in Nature Communications in its present form.

Replies to Reviewers (Revision #2)

(NCOMMS-17-09535)

Solving for Ambiguities in Radar Geophysical Exploration of Planetary Bodies by Mimicking Bats Echolocation"

L.Carrer and L. Bruzzone

First we would like to thank the anonymous Reviewers for their kind words about the paper and comments that helped to improve the quality of the manuscript. We did our best effort to provide an improved version of the manuscript in the light of the remaining Reviewer #2 comments.

According to the reviewer's suggestion, the revised manuscript includes experimental results obtained from a radargram already interpreted in the literature by planetary geologists (Putzig et al. (2014)). This radargram shows subsurface and clutter features (i.e. mixed case, intermingled using reviewer words). These new experimental results are shown in Fig.10 in the revised paper along with explanation in the results section. They confirm and point out the effectiveness of the proposed technique.

Regarding the Reviewer #2 comments on Fig. 7 and Fig. 9, we believe that the perplexities of the Reviewer were coming from the fact that we did not show the clutter simulations. Please note that we already performed, as stated in the previous version of the paper, the clutter simulations to validate our method results. Accordingly and as asked by the reviewer, we replaced panel (b) of the fore mentioned figures with the clutter simulations (i.e. clutter model derived from topography) instead of that with the digital elevation model. We understand that the clutter simulations are of more direct interpretability. The comment of Reviewer #2 on Fig. 7 regarding the presence of the subsurface is a misunderstanding because we clearly state in the paper that the example is referred to the *clutter hypothesis only* testing. Accordingly, there is no subsurface reflection in this particular example. Regarding the comment on Fig. 9, we added the clutter simulation to the figure which clearly shows that our experimental data interpretation is correct. In particular, Fig. 9 shows a mixed case result where we have clutter and subsurface in the same echo trace.

The revised paper changes are tracked in bold for the text parts and in yellow for the figures.

Reviewer #2 (Remarks to the Author):

[RC-2-1]: *In the example of clutter features for the impact craters shown in fig. 7, there is no subsurface reflector clearly distinguished from the surface clutter features.*

Author's Answer: As stated in the paper, Fig. 7 is an example of test for the *clutter only hypothesis*. Indeed, there is no subsurface reflector in that radargram. As outlined in the results section we tested three hypothesis namely (i) clutter only, (ii) signal only, and (iii) clutter plus signal (i.e mixed case). We modified Figure 7 by replacing the digital elevation model of the surface with the cluttergram (i.e. clutter model derived from the topography) (see also RC-2-2). Cluttergram has been computed with the method described in [1]. Please note that in the previous version of the manuscript, as stated in the results section and in the figure caption, we already produced the cluttergram to validate our results but we decided to show the surface digital elevation model of the acquisition scenario instead of it. As suggested by the reviewer, it is of more direct interpretation for the analysis of the results to show the cluttergram which is indeed presented in this revised version of the paper.

REFERENCES FOR THIS ANSWER:

[1] Russo, Federica, et al. "An incoherent simulator for the SHARAD experiment." *Radar Conference, 2008. RADAR'08. IEEE*. IEEE, 2008.

[RC-2-2]: *The SHARAD radargram shown in fig. 9a is confusing. Clutter features appear above the surface return in time-delay or depth. How is this possible? An example is needed, preferably a published SHARAD radargram that has intermingled subsurface reflectors and surface clutter features, where clutter features can be unambiguously attributed to surface features via ground-range projection of the radargram onto the surface or using a clutter model derived from topography.*

Author's Answer: We improved figure 9a by including the clutter simulation (i.e. the clutter model derived from topography) as the reviewer suggested. We believe that what was causing confusion is that we labeled as "Surface Reflection" the "Main Surface Reflection". Therefore the clutter features appearing above the surface return are actually surface clutter features appearing before the *main* surface return. We improved the figure legend and description to reflect this and added the cluttergram instead of the digital elevation model of the surface. We believe that the radargram we selected (Fig. 9) is an example of "intermingled" case (in the paper we call it mixed case) where we have interleaved surface and subsurface feature. This radargram is a typical example of SHARAD data product showing lineated valley fill in Deuteronilus Mensae. It has been analyzed in great detail by Jeff Plaut at a recent MARSIS/SHARAD workshop [1] (see pg. 17 to 25 of the presentation). In any case, it is clear from the clutter simulation that we have a genuine subsurface return in the analyzed radargram .

We added another example of a mixed case result. As the reviewer suggested, we took an already interpreted radargram in a recent peer-reviewed journal. In particular, we analyzed Putzig's et al. results [2] at the Phoenix Landing Site in the Northern Plains of Mars. In their work, by using SHARAD data, they highlighted and mapped the presence of a very shallow subsurface layer (see Figure [RC-2-2-1]), which has been attributed to a relatively deep base of ground ice or to ice-free sediments (e.g. lava flows). In the paper, they did extensive analysis to prove it is legitimate subsurface return and not clutter or side lobes of the surface response. Indeed, the presence of the interpreted subsurface layer is revealed and confirmed by our proposed bio-inspired method (Figure [RC-2-2-2]) along with the discrimination of a clutter return located at about 300 m equivalent depth. Please note that in their paper they revealed the subsurface interface by comparing different radargrams and different cluttergrams. Once they found a putative subsurface interface, they estimated its geoelectrical properties by time delay conversion method. This is substantially different by our proposed bio-inspired method where we can detect clutter and subsurface interface without using a clutter model derived from topography. If we compare this experiment with the previous one on Deuteronilus Mensae, in this case we tested the method for a very shallow subsurface layer. The clutter simulation shows that in principle the subsurface interface should be masked by it but this is not entirely reflected in the experimental data as shown by Putzig et. al[2]. It is reasonable to expect that residual clutter is present in the vicinity of the interface. In any case, the method proved robust enough to correctly detect the subsurface interface and discriminate it from the clutter reflection located at about 300 m depth.

REFERENCES FOR THIS ANSWER:

- [1] J. Plaut, *Lobate Debris Aprons*, SHARAD/MARSIS Data User's Workshop, 2014/03/16 in Montgomery Ballroom A at The Woodlands Marriott. Available at : http://pds-geosciences.wustl.edu/workshops/SHARAD_MARSIS_Mar14/Plaut_Deuteronilus-Mensae.pdf Page 17-25
- [2] Putzig, Nathaniel E., et al. "SHARAD soundings and surface roughness at past, present, and proposed landing sites on Mars: reflections at Phoenix may be attributable to deep ground ice." *Journal of Geophysical Research: Planets* 119.8 (2014): 1936-1949.

Figure 4. Portion of SHARAD dayside nonrolled observation 12903-01 extending from south to north across the Phoenix landing site and the Green Valley. (a) Uninterpreted radargram. (b) Interpreted radargram, showing delay times for the MOLA modeled surface in blue and for the subsurface return in yellow, with the latter delayed $\sim 0.4 \mu\text{s}$ from the surface return. "x" symbols show corresponding delay times from crossing nightside observations. Inset shows map view of ground track. Dayside nonrolled observations typically have lower signal-to-noise ratio than either nightside or rolled ones, but power of subsurface returns remains high enough to distinguish from sidelobes, and the crossing trajectories are critical for tracking features between observations.

Figure [RC-2-2-1: SHARAD Radargram analyzed by Putzig et al.[2] (Figure from the paper). Subsurface reflector in yellow in figure (b). We analyzed the same radargram. Please not that in the paper they preferred to show the radargram with a redscale color scheme. We adopt a grayscale color scheme (see Figure [RC-2-2-2])

Figure [RC-2-2-2]: Our results confirming the findings of Putzig et al.(2014). This figure has been included in the paper with an extensive caption.

[RC-2-3]: *It is still not made clear in the revision how the approach taken is not equivalent to a split-chirp technique that has been previously applied to SHARAD data.*

Author's Answer: As stated in [RC-2-2] and [RC-3-2] of the previous round of review, our proposed bio-inspired method focuses on clutter detection. According to our theoretical model, potentially we can use two very different frequencies to discriminate clutter. We tested our model on SHARAD data by splitting the band because there is no experimental sounding data with very different central frequencies (e.g.: $f_1 = 9$ MHz and $f_2 = 63$ MHz).

More importantly, split-chirp technique has been applied to MARSIS and SHARAD [1-2] merely for the estimation of dielectric properties subsurface and not for clutter discrimination whatsoever. In our case, split-chirp is just a mean for proving our theoretical model and its ability to discriminate clutter from the subsurface signal. This is a fundamental difference with what already proposed in the literature which just focus on geo-electrical properties estimation starting from very different theoretical background assumptions and final study goals. Indeed, for the geoelectrical parameter estimation, in the split-chirp case clutter is detected by using conventional simulations obtained via a digital elevation model. Moreover, our proposed method has the capability of retrieving the surface roughness which is another novel point never presented in the literature.

REFERENCES FOR THIS ANSWER:

- [1] Restano, Marco, Giovanni Picardi, and Roberto Seu. "1D-FDTD characterization of ionosphere influence on ground penetrating radar data inversion." *IEEE Transactions on Antennas and Propagation* 62.4 (2014): 2223-2230.
- [2] Iorio, M., Fois, F., Mecozzi, R., Catalo, C., Picardi, G., Seu, R., & Flamini, E. (2008, May). Mars north polar cup subsurface materials property estimation using GPR SHALLOW RADAR data. In Radar Conference, 2008. RADAR'08. IEEE (pp. 1-6). IEEE.

REVIEWERS' COMMENTS:

Reviewer #2 (Remarks to the Author):

Review of: Solving for Ambiguities in Radar Geophysical Exploration of Planetary Bodies by Mimicking Bats Echolocation (Revised)

By L. Carrer and L. Bruzzone

With the addition of the clutter simulations to Fig. 7 and 9, and the addition of Fig. 10, the revised paper is greatly improved.

I am now happy to recommend the revised paper for publication in Nature Communications.